# Integrated interfacial design of covalent organic framework photocatalysts to promote hydrogen evolution from water

Ting He[1], Wenlong Zhen[2], Yongzhi Chen[1], Yuanyuan Guo[3], Zhuoer Li[1,4], Ning Huang [5], Zhongping Li[1], Ruoyang Liu[1], Yuan Liu[1], Xu Lian[1], Can Xue[2], Tze Chien Sum [3], Wei Chen[1] & Donglin Jiang [1,4] ✉

Attempts to develop photocatalysts for hydrogen production from water usually result in low efficiency. Here we report the finding of photocatalysts by integrated interfacial design of stable covalent organic frameworks. We pre-designed and constructed different molecular interfaces by fabricating ordered or amorphous π skeletons, installing ligating or non-ligating walls and engineering hydrophobic or hydrophilic pores. This systematic interfacial control over electron transfer, active site immobilisation and water transport enables to identify their distinct roles in the photocatalytic process. The frameworks, combined ordered π skeletons, ligating walls and hydrophilic channels, work under 300–1000 nm with non-noble metal co-catalyst and achieve a hydrogen evolution rate over 11 mmol $g^{-1}$ $h^{-1}$, a quantum yield of 3.6% at 600 nm and a three-order-of-magnitude-increased turnover frequency of 18.8 $h^{-1}$ compared to those obtained with hydrophobic networks. This integrated interfacial design approach is a step towards designing solar-to-chemical energy conversion systems.

Using sustainable resources to power our development is an urgent need and a great goal. Exploiting water and sunlight—the two sustainable resources on this planet—to generate the green and clean fuel —hydrogen is highly desired. However, photocatalytic hydrogen production from water offers a promising way to produce green and clean hydrogen ($H_2$) fuels but suffers from low catalytic activity[1–3]. Progress in chemistry over the past decades has developed diverse photocatalytic systems, such as carbon nitrides[4], linear conjugated polymers[5], conjugated microporous polymers[6,7] and covalent organic frameworks (COFs)[8–11] for $H_2$ evolution. However, these systems are very limited in the catalytic activity and deeply relied on the use of noble metal platinum (Pt) co-catalyst, which restricts their further

development and application[12–23]. How to address these bottleneck issues remains a substantial challenge.

COFs are a class of crystalline porous polymers with periodically ordered π skeletons and open channels[6,24–27]. Their π skeletons could be constructed with dense chromophores to create efficient light-harvesting antennae, while their walls could be installed to ligate non-noble metal reaction centres and the pores could be engineered to facilitate the delivery of water molecules. However, a systematic interfacial design of photocatalysts is unprecedented.

In this work, we explored a strategy for integrated interfacial designs to differentiate molecular interfaces that control electron transfer, active centre immobilisation and water transport, with an aim

[1]Department of Chemistry, Faulty of Science, National University of Singapore, 3 Science Drive 3, Singapore 117543, Singapore. [2]School of Materials Science and Engineering, Nanyang Technological University, 50 Nanyang Avenue, Singapore 639798, Singapore. [3]Division of Physics and Applied Physics, School of Physical and Mathematical Sciences, Nanyang Technological University, 21 Nanyang Link, Singapore 637371, Singapore. [4]Joint School of National University of Singapore and Tianjin University, International Campus of Tianjin University, Binhai New City, Fuzhou 350207, China. [5]MOE Key Laboratory of Macromolecular Synthesis and Functionalisation, Department of Polymer Science and Engineering, Zhejiang University, Hangzhou 310027, China. ✉e-mail: chmjd@nus.edu.sg

to develop efficient and robust photocatalysts to enable H$_2$ production from water. We found that photocatalysts that merge ordered π skeletons, ligating walls and hydrophilic channels can achieve a H$_2$ evolution rate over 11 mmol g$^{-1}$ h$^{-1}$, a quantum yield of 3.6% at 600 nm and a turnover frequency of 18.8 h$^{-1}$ which is increased by more than three orders of magnitude from the results obtained with hydrophobic networks. Remarkably, the photocatalyst is stable to work with non-noble metal co-catalyst under a wide range of light from 300 nm to 1000 nm. These insights open the way to actionable photocatalysts for green fuel production.

## Results

### Design principle

Photocatalytic hydrogen generation from water involves a series of continuous photochemical events from light harvesting to charge separation and electron transfer, as well as the water delivery to the catalytic centres. How to merge these physical and chemical processes into one material in a seamless manner is key to hydrogen evolution. To address this fundamental key issue, we predesigned and constructed π skeletons to be ordered or amorphous, walls to be ligating or non-ligating and pores to be hydrophilic or hydrophilic, to constitute different combinations of molecular interfaces into the photocatalysts (Fig. 1a). The combination of these different molecular interfaces leads to the generation of a series of photocatalysts that possess distinct components and structures (Fig. 1b). This systematic interfacial design enables to identify their roles in the photocatalytic process and leads to the finding of efficient photocatalysts and new insights on photo-to-chemical energy conversion.

### Synthesis

We selected porphyrin as the knot to construct π-conjugated COFs to combine light-harvesting efficiency extended to over 1000 nm with exciton migration capability (Fig. 1b). Pyrazine linker component was designed to immobilise the non-noble metal [Mo$_3$S$_{13}$]$^{2-}$ reaction centre to form catalytic pore walls[28], while the dihydroxyphenyl unit was engineered as another linker component to regulate the hydrophilicity of pores that controls water access to the Mo catalytic sites. The ZnP-Pz-DHTP-COF (Fig. 1c) was synthesised by the three-component polycondensation of zinc 5,10,15,20-tetrakis(p-tetraphenylamino)porphyrin (ZnP), pyrazine-2,5-dicarbaldehyde (PzDA) and 2,5-dihydroxyterephthalaldehyde (DHTA) with a molar ratio of ZnP/PzDA/DHTA = 1/1/1 under solvothermal conditions in 86% yield. The $^1$H NMR spectroscopy of hydrolysed ZnP-Pz-DHTP-COF revealed that the molar ratio of PzDA and DHTA is 1/1, confirming the successful synthesis of three-component COF (Supplementary Fig. 1). The hydrophobic ZnP-Pz-COF (Fig. 1b) was prepared by the polymerisation of ZnP with PzDA (molar ratio = 1/2) in 88% yield. Amorphous ZnP-Pz-DHTP-POP (POP = porous organic polymer), as a non-crystalline control of ZnP-Pz-DHTP-COF, was prepared in 81% yield (Fig. 1b). By replacing pyrazine with phenyl linker, ZnP-TP-DHTP-COF without pyrazine ligating units on walls was synthesised as a control in 85% yield.

The hydrophilic ZnP-Pz-PEO-COF with pyrazine ligating sites was synthesised in 96% yield by the reaction of ZnP-Pz-DHTP-COF with 1-bromo-2-(2-methoxyethoxy)ethane in the presence of K$_2$CO$_3$ in DMF at 85 °C (Fig. 1c–e). Similarly, ZnP-Pz-PEO-POP and ZnP-TP-PEO-COF were prepared in 98% and 96% yields by reacting ZnP-Pz-DHTP-POP and

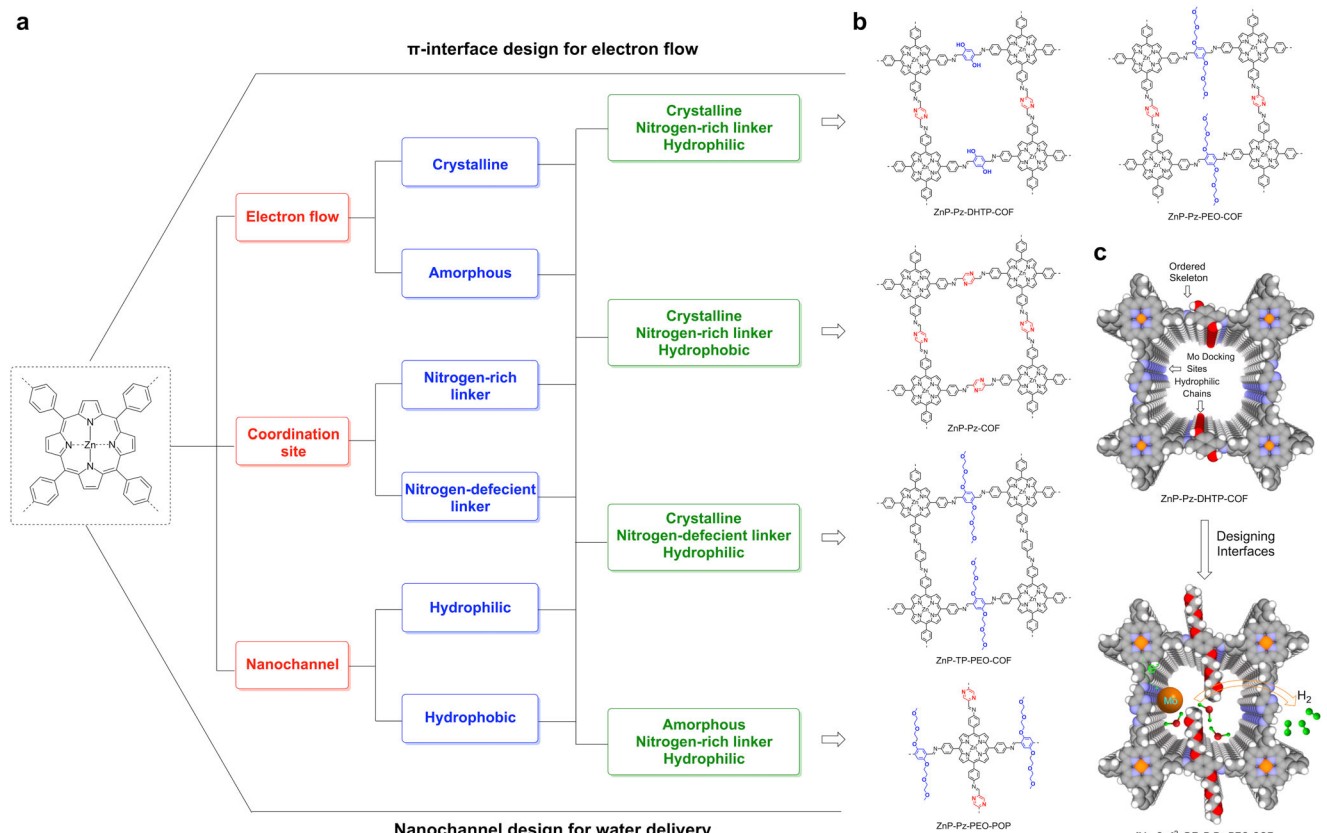

**Fig. 1 | Integrated interfacial designs of photocatalysts. a** The integrated interfacial design strategy for constructing π skeleton, wall and pore to merge three different interfaces, i.e., electron flow, active site ligating and water transport, into the photocatalysts to promote photocatalytic H$_2$ production from water. **b** Schematics of ZnP-Pz-DHTP-COF, ZnP-Pz-PEO-COF, ZnP-Pz-COF, ZnP-TP-PEO-COF and ZnP-Pz-PEO-POP; they possess built-in interfaces that are distinct from each other. **c** Schematic of merging three different interfaces into one framework photocatalyst for H$_2$ evolution.

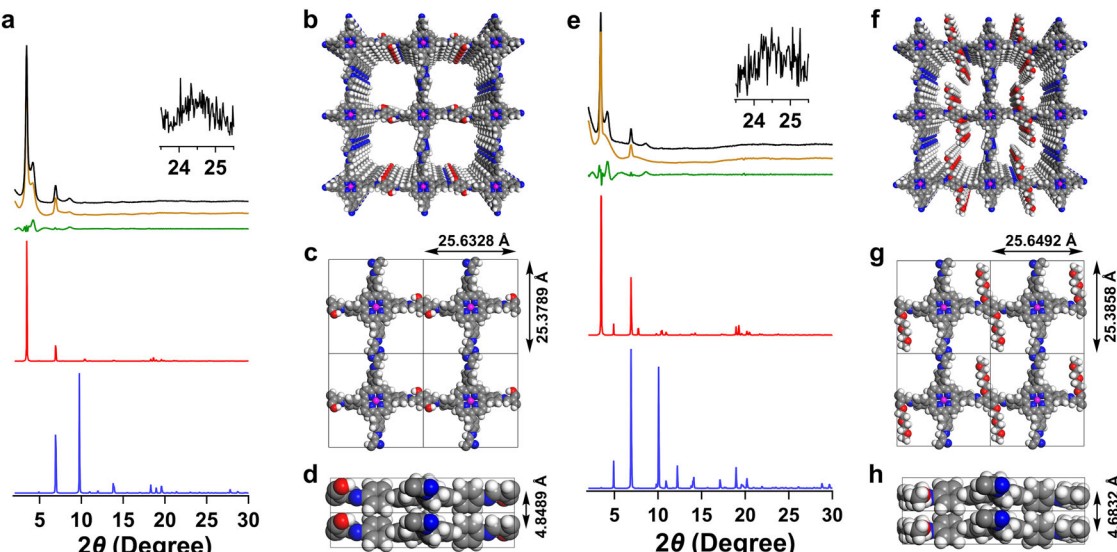

**Fig. 2 | PXRD pattern and crystal structure. a** PXRD profiles of ZnP-Pz-DHTP-COF (black, experimentally observed; orange, Pawley refined; green, their difference; red, AA-stacking mode; blue, AB-stacking mode). **b**–**d**, Reconstructed crystal structure of ZnP-Pz-DHTP-COF with (**b**) ten layers, (**c**) one layer and (**d**) two layers. **e** PXRD profiles of ZnP-Pz-PEO-COF (black, experimentally observed; orange, Pawley refined; green, their difference; red, AA-stacking mode; blue, AB-stacking mode). **f**–**h** Reconstructed crystal structure of ZnP-Pz-PEO-COF with (**f**) ten layers, (**g**) one layer, and (**h**) two layers.

ZnP-TP-DHTP-COF with 1-bromo-2-(2-methoxyethoxy)ethane, respectively (Fig. 1c).

The thiomolybdate $(NH_4)_2Mo_3S_{13}$ was synthesised by refluxing $(NH_4)_6Mo_7O_{24}\cdot4H_2O$ in an ammonium polysulfide solution[28]. The powder X-ray diffraction (PXRD) curve of $(NH_4)_2Mo_3S_{13}$ (Supplementary Fig. 2) was the same as reported in literature[28], which confirmed the same crystal structure. With this result, $[Mo_3S_{13}]^{2-}$@ZnP-Pz-COF, $[Mo_3S_{13}]^{2-}$@ZnP-Pz-DHTP-COF, $[Mo_3S_{13}]^{2-}$@ZnP-Pz-PEO-COF, $[Mo_3S_{13}]^{2-}$@ZnP-Pz-PEO-POP and $[Mo_3S_{13}]^{2-}$/ZnP-TP-PEO-COF were prepared by reacting ZnP-Pz-COF, ZnP-Pz-DHTP-COF, ZnP-Pz-PEO-COF, ZnP-Pz-PEO-POP and ZnP-TP-PEO-COF with $(NH_4)_2Mo_3S_{13}$ in methanol and isolated in quantitative yields.

## Crystalline structures

The PXRD pattern of ZnP-Pz-DHTP-COF (Fig. 2a, black curve) revealed strong peaks at 3.4°, 6.8°, 10.2° and 24.5°, which were assigned to the (100), (200), (300) and (001) facets, respectively. The ZnP-Pz-DHTP-COF adopted an AA-stacking mode (red curve), which reproduced the experimental PXRD pattern. In contrast, the AB-stacking mode (blue curve) cannot reproduce the PXRD pattern. Pawley refinement (orange curve) showed negligible difference (green curve) from the experimental profile (Supplementary Table 1). The ZnP-Pz-DHTP-COF (Fig. 2b–d) adopts a $P4/M$ lattice with $a = 25.6328$ Å, $b = 25.3789$ Å, $c = 4.8489$ Å and $\alpha = \beta = \gamma = 90°$ (for atomistic coordinates see Supplementary Table 2). The ZnP-Pz-PEO-COF (Fig. 2e, black curve) with bis(2-methoxyethyl)ether chains displayed the PXRD peaks at 3.4°, 6.8°, 8.6° and 24.5°, which were attributed to the (100), (200), (210) and (001) facets, respectively. The AA-stacking mode (red curve) reproduced the PXRD pattern, while the AB-stacking mode (blue curve) showed a greatly deviated PXRD curve. The Pawley refinement (orange curve; Supplementary Table 1) confirmed the correctness of peak assignment with small difference (green curve). The small peak at 4.0° might be attributed to the formation of slipped *J*-aggregate stacks, which was also observed for other porphyrin-based COFs[29]. The ZnP-Pz-PEO-COF (Fig. 2f–h; for atomistic coordinates see Supplementary Table 3) also adopts a $P4/M$ lattice with parameters of $a = 25.6492$ Å, $b = 25.3858$ Å, $c = 4.6832$ Å and $\alpha = \beta = \gamma = 90°$. This observation indicates that the crystal structure of ZnP-Pz-DHTP-COF is well retained in ZnP-Pz-PEO-COF. The ZnP-Pz-COF (Supplementary Fig. 3a), ZnP-TP-DHTP-COF

(Supplementary Fig. 3b) and ZnP-TP-PEO-COF (Supplementary Fig. 3c) exhibited similar crystal structures to their corresponding ZnP-Pz-DHTP-COF and ZnP-Pz-PEO-COF, respectively. The $[Mo_3S_{13}]^{2-}$@ZnP-Pz-PEO-COF retained the crystal structure of ZnP-Pz-PEO-COF after ligating with $[Mo_3S_{13}]^{2-}$, while the minor peaks at 10.5°, 16.3°, 24.5° and 27.3° (Fig. 3a, peaks marked with red stars) originated from $[Mo_3S_{13}]^{2-}$. Meanwhile, $[Mo_3S_{13}]^{2-}$/ZnP-TP-PEO-COF (Supplementary Fig. 3f) and $[Mo_3S_{13}]^{2-}$@ZnP-Pz-PEO-POP (Supplementary Fig. 3g) also exhibited the PXRD peaks of $[Mo_3S_{13}]^{2-}$.

## Characterisation

The appearance of C=N stretching band at 1612 $cm^{-1}$ in Fourier transform infrared (FTIR) spectra (Supplementary Fig. 4a–d) confirmed the successful formation of imine bonds for ZnP-Pz-COF, ZnP-Pz-DHTP-COF, ZnP-TP-DHTP-COF and ZnP-Pz-DHTP-POP. Compared to ZnP-Pz-DHTP-COF (Fig. 3b, black curve), FTIR profile of ZnP-Pz-PEO-COF (red curve) revealed two new peaks at 2874 and 1100 $cm^{-1}$, which were assigned to the alkyl C–H and C–O–C stretching bands, respectively, indicating the successful introduction of hydrophilic bis(2-methoxyethyl)ether chains. These new bands were also observed in ZnP-TP-PEO-COF (Supplementary Fig. 4e) and ZnP-Pz-PEO-POP (Supplementary Fig. 4f).

Compared to ZnP-Pz-DHTP-COF (Fig. 3c, black curve), ZnP-Pz-PEO-COF (red curve) displayed new peaks at 58 and 71 ppm in the solid-state $^{13}C$ cross polarisation magic angle spinning nuclear magnetic resonance (CP/MAS NMR) spectra, which were assigned to the methyl and methylene groups of bis(2-methoxyethyl)ether chains, respectively, again confirming the introduction of hydrophilic chains to the pore walls. The content of Zn in ZnP-Pz-DHTP-COF was determined to be 6.54 wt% by inductively coupled plasma optical emission spectroscopy, which was close to the theoretical value (6.62 wt%), revealing the stability of zinc porphyrin species under the solvothermal conditions. Elemental analysis was conducted to investigate the content of bis(2-methoxyethyl)ether chains. The relative content of C, H and N decreased from 64.27%, 3.56% and 12.88% to 58.61%, 3.32% and 10.91%, respectively, indicating that 93% hydroxyl groups of ZnP-Pz-DHTP-COF were transformed to bis(2-methoxyethyl)ether chains. Thermogravimetric analysis (TGA) under $N_2$ revealed a thermal stability over 400 °C for these COF and POP samples (Supplementary Fig. 5).

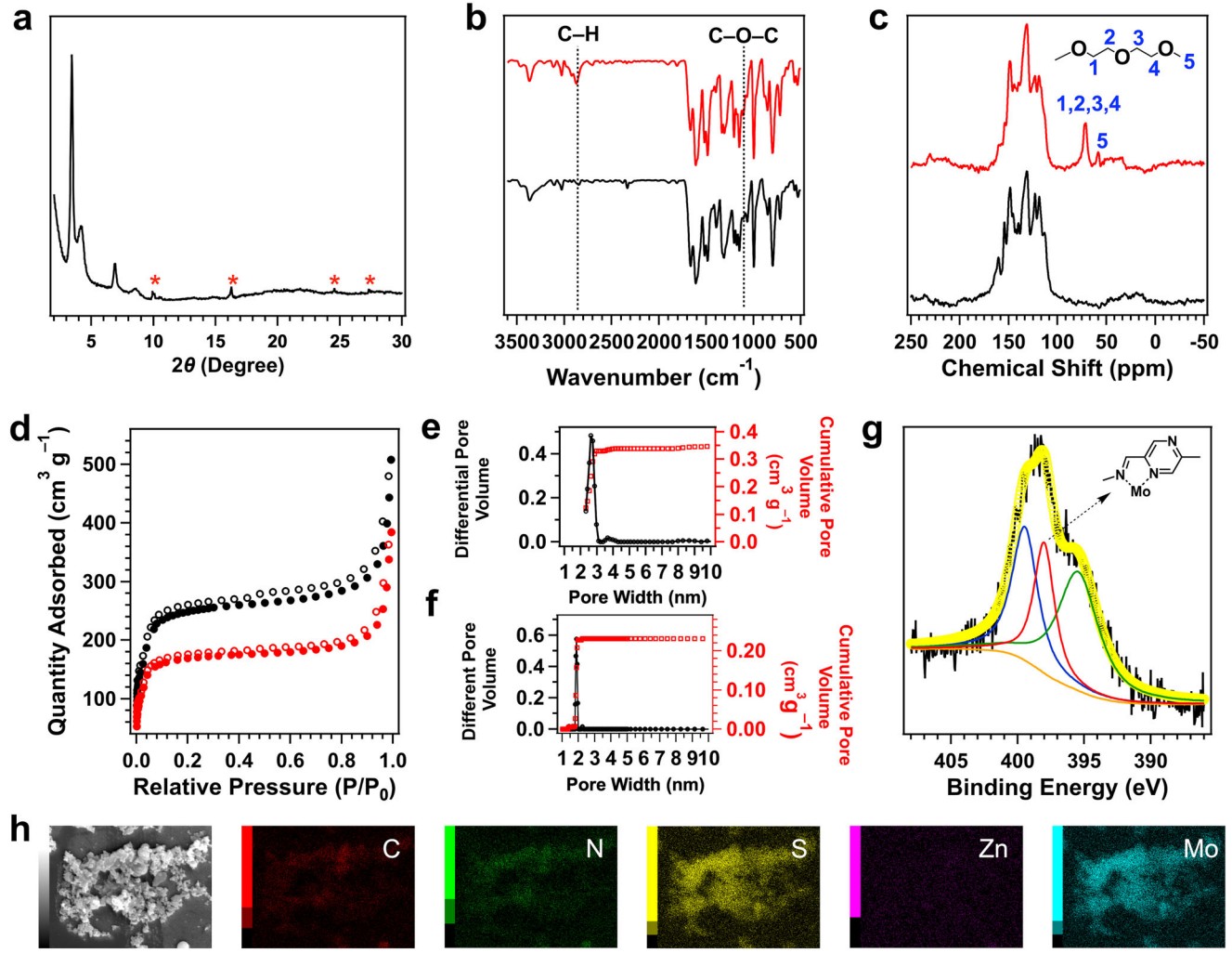

**Fig. 3 | Characterisation. a** PXRD pattern of $[Mo_3S_{13}]^{2-}$@ZnP-Pz-PEO-COF (peaks with red star due to $[Mo_3S_{13}]^{2-}$). **b** FTIR spectra of ZnP-Pz-DHTP-COF (black curve) and ZnP-Pz-PEO-COF (red curve). **c** The solid-state $^{13}$C CP/MAS NMR spectra of ZnP-Pz-DHTP-COF (black curve) and ZnP-Pz-PEO-COF (red curve). **d** Nitrogen sorption isotherm curves of ZnP-Pz-DHTP-COF (black circles) and ZnP-Pz-PEO-COF (red circles). **e** Pore size and distribution profiles of ZnP-Pz-DHTP-COF. **f** Pore size and distribution profiles of ZnP-Pz-PEO-COF. **g** XPS N 1s spectrum of $[Mo_3S_{13}]^{2-}$@ZnP-Pz-PEO-COF (Inset: bonding between pyrazine wall and $[Mo_3S_{13}]^{2-}$.). **h** EDX elemental mapping of $[Mo_3S_{13}]^{2-}$@ZnP-Pz-PEO-COF.

Scanning electron microscopy (Supplementary Fig. 6) images revealed a particle-like morphology.

The $[Mo_3S_{13}]^{2-}$@ZnP-Pz-PEO-COF was further investigated by X-ray photoelectron spectroscopy (XPS), which revealed the composition of C, N, O, Zn, Mo and S elements (Supplementary Fig. 7a). Deconvolution of the N 1s region by peak fitting disclosed a new single peak at 398.0 eV (Fig. 3g, red curve) compared to those of ZnP-Pz-PEO-COF and $(NH_4)_2Mo_3S_{13}$ (Supplementary Fig. 7b, c), which originates from the Mo–N bond between $[Mo_3S_{13}]^{2-}$ and N atoms of imine-pyrazine units in a five-membered structure[30] (Fig. 3g, inset). The Mo 3d spectrum (Supplementary Fig. 7d) was deconvoluted into two peaks at 230.0 and 233.0 eV, which were assigned to Mo $3d_{5/2}$ and Mo $3d_{3/2}$, respectively[28]. The N 1s spectrum of $[Mo_3S_{13}]^{2-}$/ZnP-TP-PEO-COF (Supplementary Fig. 7e) was deconvoluted into two peaks at 395.2 and 398.5 eV, which were assigned to $(NH_4)_2Mo_3S_{13}$ and ZnP-TP-PEO-COF, respectively. These results indicate the necessity of pyrazine unit for the Mo–N bond formation to ligate $[Mo_3S_{13}]^{2-}$ to the pore walls in $[Mo_3S_{13}]^{2-}$@ZnP-Pz-PEO-COF.

Elemental mapping with energy-dispersive X-ray spectroscopy (EDX) demonstrated that the elements of C, N, Zn, Mo and S were uniformly dispersed in the $[Mo_3S_{13}]^{2-}$@ZnP-Pz-PEO-COF particle (Fig. 3h). The Mo content in $[Mo_3S_{13}]^{2-}$@ZnP-Pz-PEO-COF, $[Mo_3S_{13}]^{2-}$@ZnP-Pz-DHTP-COF, $[Mo_3S_{13}]^{2-}$@ZnP-Pz-PEO-POP and $[Mo_3S_{13}]^{2-}$/ZnP-TP-PEO-COF were determined to be 5.63 wt%, 6.21 wt%, 5.55 wt%, 5.48 wt% and 4.94 wt%, respectively, by inductively coupled plasma – optical emission spectrometry (Supplementary Table 4).

## Porosity studies

The ZnP-Pz-DHTP-COF displayed a typical type IV nitrogen sorption isotherm, suggesting a mesoporous character (Fig. 3d, black circles). The Brunauer–Emmett–Teller (BET) surface area was calculated to be 790 m² g⁻¹. The pore size distribution curves (Fig. 3e) revealed that ZnP-Pz-DHTP-COF has a pore size of 2.6 nm and a pore volume of 0.35 cm³ g⁻¹. The BET surface area, pore size and pore volume of ZnP-Pz-PEO-COF decreased to be 557 m² g⁻¹, 1.9 nm and 0.23 cm³ g⁻¹ (Fig. 3d, red circles, Fig. 3f), respectively, owing to the occupation of pores by the hydrophilic chains. The ZnP-Pz-COF (Supplementary Fig. 8a, b), ZnP-TP-DHTP-COF (Supplementary Fig. 8c, d) and ZnP-Pz-DHTP-POP (Supplementary Fig. 8e, f) exhibited the same porous characters with the BET surface area of 658, 634 and 388 m² g⁻¹, and a pore volume of 0.31 0.26 and 0.16 cm³ g⁻¹, respectively. After integration with bis(2-methoxyethyl) ether chains, the BET surface area of ZnP-TP-PEO-COF (Supplementary Fig. 8g, h) and ZnP-Pz-PEO-POP (Supplementary Fig. 8i, j)

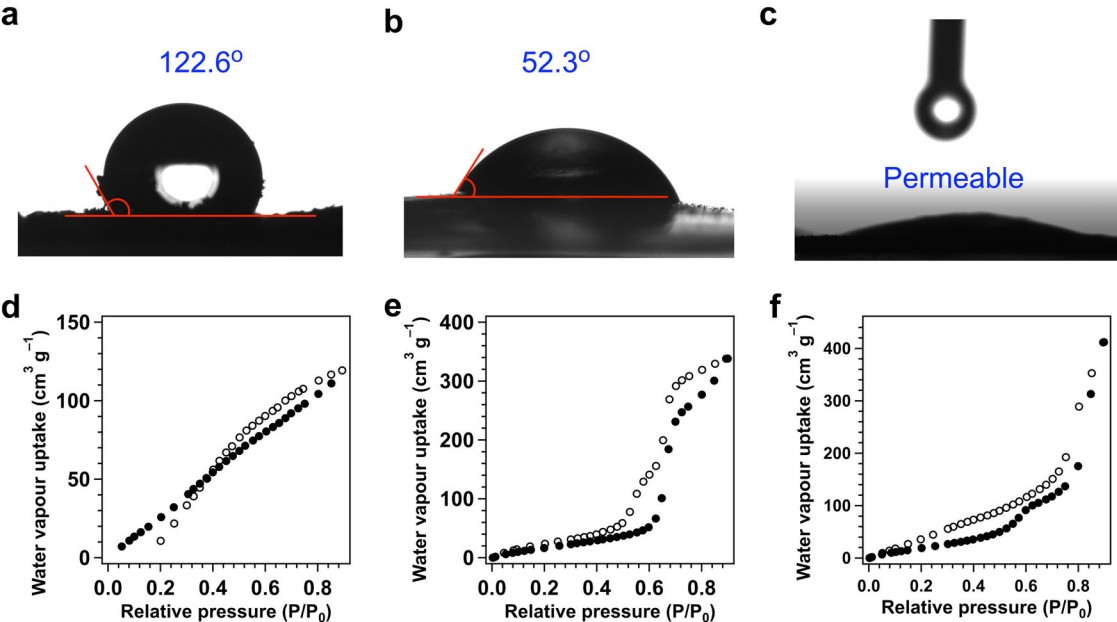

**Fig. 4 | Hydrophobicity and water uptake. a–c** Images of contact angle measurements of (**a**) ZnP-Pz-COF, (**b**) ZnP-Pz-DHTP-COF and (**c**) ZnP-Pz-PEO-COF. **d–f** Water vapour sorption isotherms of (**d**) ZnP-Pz-COF, (**e**) ZnP-Pz-DHTP-COF and (**f**) ZnP-Pz-PEO-COF measured at 298 K.

decreased to 474 and 171 m$^2$ g$^{-1}$, while the pore volume decreased to 0.19 and 0.05 cm$^3$ g$^{-1}$, respectively.

The [Mo$_3$S$_{13}$]$^{2-}$@ZnP-Pz-PEO-COF exhibited the BET surface area (Supplementary Fig. 8k) and pore volume (Supplementary Fig. 8l) of 362 m$^2$ g$^{-1}$ and 0.16 cm$^3$ g$^{-1}$, respectively. Meanwhile, a new pore of 1.5 nm was generated (Supplementary Fig. 8l). The BET surface area of [Mo$_3$S$_{13}$]$^{2-}$/ZnP-TP-PEO-COF and [Mo$_3$S$_{13}$]$^{2-}$@ZnP-Pz-PEO-POP (Supplementary Fig. 8m, o) was 248 and 112 m$^2$ g$^{-1}$, while the pore volume (Supplementary Fig. 8n and p) was 0.12 and 0.05 cm$^3$ g$^{-1}$, respectively.

### Hydrophilicity and contact angle

As expected, ZnP-Pz-COF is hydrophobic to exhibit a contact angle of 122.6° (Fig. 4a). The contact angle decreased to 52.3° (Fig. 4b) for ZnP-Pz-DHTP-COF. More explicitly, ZnP-Pz-PEO-COF exhibited super hydrophilicity and cannot be screened to give an image of contact angle as water permeated through the COF films instantly (Fig. 4c, Supplementary Movie 1). The [Mo$_3$S$_{13}$]$^{2-}$@ZnP-Pz-COF, [Mo$_3$S$_{13}$]$^{2-}$@ZnP-Pz-DHTP-COF and [Mo$_3$S$_{13}$]$^{2-}$@ZnP-Pz-PEO-COF exhibited similar hydrophilic tendency to with pristine COFs (Supplementary Fig. 9, Supplementary Moviex 2). Moreover, we conducted water uptake experiments at 298 K. The maximum water uptake of ZnP-Pz-COF (Fig. 4d), ZnP-Pz-DHTP-COF (Fig. 4e) and ZnP-Pz-PEO-COF (Fig. 4f) is 119, 338 and 412 cm$^3$ g$^{-1}$ at $P/P_0$ = 0.9, respectively, which is consistent with the contact angle result. The hydrophilic nanochannel interface promotes the water transport to the catalytic sites.

### Band gap structure

The electronic diffuse reflection absorption spectra were investigated to reveal light absorption and optical band gap. All samples exhibited a broad and strong absorption band from 300 nm to over 1000 nm in the visible and near-infrared regions with a maximum absorbance at 612 nm, demonstrating a high light-harvesting efficiency (Fig. 5a). The ZnP-Pz-DHTP-COF (Fig. 5a, red curve) displayed a slightly blue-shifted absorption edge at 706 nm compared to ZnP-Pz-COF at 733 nm (Fig. 5a, black curve). The ZnP-Pz-PEO-COF exhibited a red-shifted absorption edge at 739 nm (Fig. 5a, blue curve). The ZnP-TP-PEO-COF (Fig. 5a, green curve) presented a negligible change while ZnP-Pz-PEO-POP (Fig. 5a, purple curve) showed a red-shifted absorption edge at 764 nm compared to that of ZnP-Pz-PEO-COF.

By using Kubelka−Munk function, the optical band gap of ZnP-Pz-COF, ZnP-Pz-DHTP-COF, ZnP-Pz-PEO-COF, ZnP-TP-PEO-COF and ZnP-Pz-PEO-POP was estimated to be 1.49, 1.51, 1.45, 1.46 and 1.44 eV (Supplementary Fig. 10a and Supplementary Table 5), respectively. The (NH$_4$)$_2$Mo$_3$S$_{13}$ exhibited an absorption band from 350 nm to 700 nm with a maximum absorbance at 535 nm (Supplementary Fig. 10b). The optical band gap of [Mo$_3$S$_{13}$]$^{2-}$@ZnP-Pz-COF, [Mo$_3$S$_{13}$]$^{2-}$@ZnP-Pz-DHTP-COF, [Mo$_3$S$_{13}$]$^{2-}$@ZnP-Pz-PEO-COF, [Mo$_3$S$_{13}$]$^{2-}$/ZnP-TP-PEO-COF and [Mo$_3$S$_{13}$]$^{2-}$@ZnP-Pz-PEO-POP was evaluated to be 1.51, 1.47, 1.46, 1.48 and 1.43 eV (Supplementary Fig. 10c, Supplementary Table 2), respectively. These results indicate that these materials are low band gap photocatalysts.

Cyclic voltammetry experiments (Supplementary Figs. 11 and 12) were conducted to determine the highest occupied molecular orbital (HOMO) and lowest unoccupied molecular orbital (LUMO) (Fig. 5b and Supplementary Fig. 13). The HOMO level of ZnP-Pz-COF, ZnP-Pz-DHTP-COF, ZnP-Pz-PEO-COF, ZnP-TP-PEO-COF and ZnP-Pz-PEO-POP was evaluated to be −5.25, −5.71, −5.63, −5.71 and −5.34, while the LUMO level was −3.81, −3.83, −3.87, −3.90 and −3.88 eV, respectively (Fig. 5b, Supplementary Table 5). The introduction of hydroxyl groups and bis(2-methoxyethyl)ether chains led to the decrease of LUMO level of COFs from −3.81 to −3.83 and −3.87 eV. Clearly, the LUMO levels of these COFs and POP are more negative than the redox potential for water reduction (−4.02 eV versus the vacuum level), thus enabling hydrogen evolution from water.

Upon integration of [Mo$_3$S$_{13}$]$^{2-}$, the resulting photocatalysts did not show obvious changes in their electronic band structures. The HOMO level of [Mo$_3$S$_{13}$]$^{2-}$@ZnP-Pz-COF, [Mo$_3$S$_{13}$]$^{2-}$@ZnP-Pz-DHTP-COF, [Mo$_3$S$_{13}$]$^{2-}$@ZnP-Pz-PEO-COF, [Mo$_3$S$_{13}$]$^{2-}$/ZnP-TP-PEO-COF and [Mo$_3$S$_{13}$]$^{2-}$@ZnP-Pz-PEO-POP was calculated to be −5.25, −5.73, −5.68, −5.64 and −5.24, while their LUMO level was −3.87, −3.86, −3.87, −3.95 and −3.68 eV, respectively (Supplementary Fig. 13 and Supplementary Table 2).

### Photocatalytic activity

Photocatalytic hydrogen evolution experiments were conducted in systems with [Mo$_3$S$_{13}$]$^{2-}$@ZnP-Pz-PEO-COF catalyst (10 mg) and sacrificial donor in water (10 mL), upon irradiation with a 300 W Xenon lamp ($\lambda > 420$ nm). Firstly, ascorbic acid (50 mM) was used as a

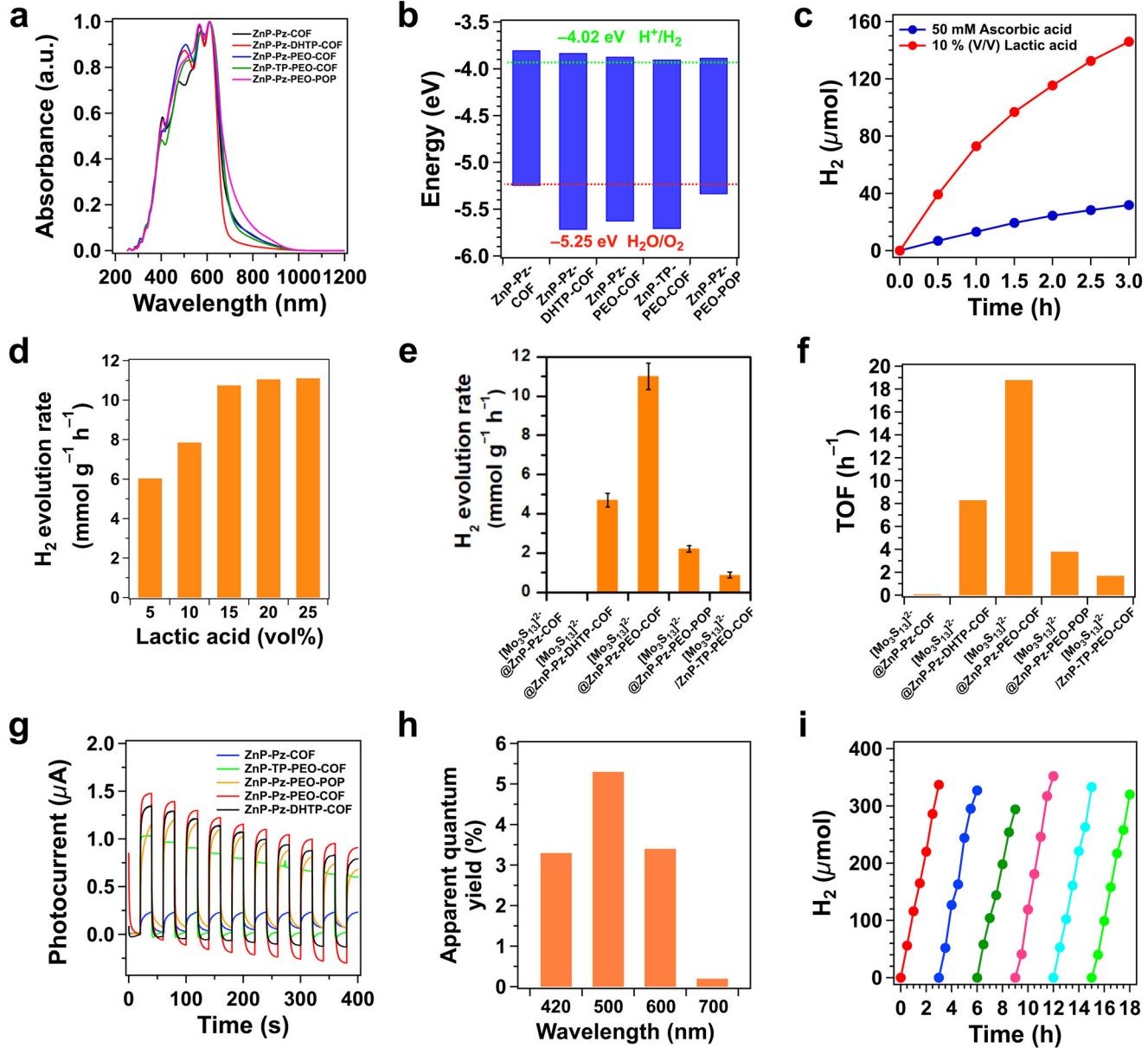

**Fig. 5 | Optoelectronic property and photocatalytic activity. a** Solid-state electronic diffuse reflection absorption spectra of ZnP-Pz-COF (black), ZnP-Pz-DHTP-COF (red), ZnP-Pz-PEO-COF (blue), ZnP-TP-PEO-COF (green) and ZnP-Pz-PEO-POP (purple). **b** Band gap structures of ZnP-Pz-COF, ZnP-Pz-DHTP-COF, ZnP-Pz-PEO-COF, ZnP-TP-PEO-COF and ZnP-Pz-PEO-POP as well as band energy for proton reduction. **c**, Time-dependent hydrogen production profiles monitored over 3 h with $[Mo_3S_{13}]^{2-}$@ZnP-Pz-PEO-COF in the presence of 50 mM ascorbic acid (red curve) and 10 vol% lactic acid (blue curve). **d** Hydrogen evolution rates of $[Mo_3S_{13}]^{2-}$@ZnP-Pz-PEO-COF with different lactic acid concentrations. **e** Hydrogen evolution rates of $[Mo_3S_{13}]^{2-}$@ZnP-Pz-PEO-COF compared with other analogues (error bars are calculated based on three independent experiments). **f** TOF of $[Mo_3S_{13}]^{2-}$@ZnP-Pz-COF, $[Mo_3S_{13}]^{2-}$@ZnP-Pz-DHTP-COF, $[Mo_3S_{13}]^{2-}$@ZnP-Pz-PEO-COF, $[Mo_3S_{13}]^{2-}$@ZnP-Pz-PEO-POP and $[Mo_3S_{13}]^{2-}$/ZnP-TP-PEO-COF. **g** Photocurrent generation of different samples coated on an indium-tin-oxide electrode as a working electrode upon light on-off switch. **h** Apparent quantum yields of $[Mo_3S_{13}]^{2-}$@ZnP-Pz-PEO-COF under monochromatic lights of different wavelengths. **i** Stability of $[Mo_3S_{13}]^{2-}$@ZnP-Pz-PEO-COF upon six-cycle photocatalytic operation under continuous irradiation ($\lambda > 420$ nm).

sacrificial reagent, $H_2$ was generated smoothly at a rate of 1.38 mmol g$^{-1}$ h$^{-1}$ (Fig. 5c, blue curve). Impressively, the $H_2$ evolution rate greatly increased to 7.8 mmol g$^{-1}$ h$^{-1}$ when ascorbic acid was replaced with lactic acid (10 vol%) (Fig. 5c, red curve). Moreover, different lactic acid contents of 5 vol%, 15 vol% and 20 vol% were investigated to evaluate the $H_2$ production. As a result, a lactic acid concentration of 15 vol% yielded the highest $H_2$ evolution rate (Fig. 5d). To exclude the effect of pH values and concentrations between ascorbic acid and lactic acid, we adjusted the pH value of 1.17 M ascorbic acid to 1.46 with diluted HCl aqueous solution, which presented the same concentration and pH value of 15 vol% lactic acid. The resultant $H_2$ evolution rate was only 1.41 mmol g$^{-1}$ h$^{-1}$ (Supplementary Figs. 14 and 15), revealing the better

performance of lactic acid. Interestingly, the $H_2$ production rate demonstrated a volcano-like trend related to the pH value of lactic acid aqueous solution, with the highest rate of 14.5 mmol g$^{-1}$ h$^{-1}$ (Supplementary Figs. 14 and 15) over 15 vol% lactic acid with pH value of 4. We further tuned the content of $[Mo_3S_{13}]^{2-}$ (Supplementary Table 1) and found that $[Mo_3S_{13}]^{2-}$@ZnP-Pz-PEO-COF containing 5.61 wt% Mo achieved the highest rate of 11 mmol g$^{-1}$ h$^{-1}$ (Supplementary Figs. 16a and 17).

Remarkably, $[Mo_3S_{13}]^{2-}$@ZnP-Pz-PEO-COF is comparable to or even higher than those of the state-of-the-art Pt-based systems (Supplementary Table 6). Moreover, $[Mo_3S_{13}]^{2-}$@ZnP-Pz-PEO-COF is much far superior in the rate which is two orders of magnitude as high as

those of reported non-noble metal-based systems. For example, the Co-based $N_2$-COF and COF-42 only work in $CH_3CN/H_2O$ (4/1 in vol) to show a rate of only 0.782 and 0.163 mmol g$^{-1}$ h$^{-1}$, respectively[9,11], while the Ni-based TpDTz COF[10] and Mo-based EB-COF[31] in water result in a rate of 0.941 mmol g$^{-1}$ h$^{-1}$ and less than 2 mmol g$^{-1}$ h$^{-1}$, respectively.

To elucidate the origin of exceptional efficiency of [Mo$_3$S$_{13}$]$^{2-}$@ZnP-Pz-PEO-COF, we investigated the role of two interfaces in the $H_2$ evolution, i.e., the hydrophilic pore interface for water transport and ordered ligating skeleton interface for electron flow. The $H_2$ evolution rate of hydrophilic [Mo$_3$S$_{13}$]$^{2-}$@ZnP-Pz-PEO-COF is 2.3- and 1100-fold higher than those of medium-hydrophilic [Mo$_3$S$_{13}$]$^{2-}$@ZnP-Pz-DHTP-COF (4.7 mmol g$^{-1}$ h$^{-1}$) and hydrophobic [Mo$_3$S$_{13}$]$^{2-}$@ZnP-Pz-COF (10.1 μmol g$^{-1}$ h$^{-1}$), respectively (Fig. 5e and Supplementary Fig. 16b). With the hydrophilic pores, water molecules can be easily delivered to the catalytic sites, leading to a high $H_2$ evolution rate. Notably, this hydrophilic interface greatly improves the utility of Mo reaction centre. Indeed, the hydrogen evolution rate based on Mo metal in [Mo$_3$S$_{13}$]$^{2-}$@ZnP-Pz-PEO-COF was 196 mmol g(Mo)$^{-1}$ h$^{-1}$, which was also 1200-fold as high as that of hydrophobic [Mo$_3$S$_{13}$]$^{2-}$@ZnP-Pz-COF.

We further investigated the electron transfer interface by comparing with [Mo$_3$S$_{13}$]$^{2-}$/ZnP-TP-PEO-COF. Dramatically, the $H_2$ evolution rate of non-ligating [Mo$_3$S$_{13}$]$^{2-}$/ZnP-TP-PEO-COF decreased to only 0.87 mmol g$^{-1}$ h$^{-1}$, which is even 12 times as low as that of ligating [Mo$_3$S$_{13}$]$^{2-}$@ZnP-Pz-PEO-COF (Fig. 5e and Supplementary Fig. 16b). This result indicates that immobilisation of [Mo$_3$S$_{13}$]$^{2-}$ onto pore walls facilitates electron transfer from the framework light-harvesting antennae to the Mo reaction centres.

We investigated the effect of crystalline skeletons by comparison with [Mo$_3$S$_{13}$]$^{2-}$@ZnP-Pz-PEO-POP bearing an amorphous π skeleton. The amorphous [Mo$_3$S$_{13}$]$^{2-}$@ZnP-Pz-PEO-POP exhibited a much lower $H_2$ evolution rate of 2.2 mmol g$^{-1}$ h$^{-1}$ (Fig. 5e and Supplementary Fig. 16b), which is five times as low as that of crystalline [Mo$_3$S$_{13}$]$^{2-}$@ZnP-Pz-PEO-COF. The low activity of [Mo$_3$S$_{13}$]$^{2-}$@ZnP-Pz-PEO-POP is ascribed to the disordered π structures, which is unfavourite for electron transport. The crystalline π skeleton also greatly improved the utility of Mo reaction centre. Indeed, the $H_2$ evolution rate based on the Mo centre in [Mo$_3$S$_{13}$]$^{2-}$@ZnP-Pz-PEO-COF is 196 mmol g(Mo)$^{-1}$ h$^{-1}$, which is five-fold as high as that of amorphous [Mo$_3$S$_{13}$]$^{2-}$@ZnP-Pz-PEO-POP.

We used turnover frequency (TOF) based on the amount of Mo to evaluate the photocatalytic activity. Impressively, the TOF of [Mo$_3$S$_{13}$]$^{2-}$@ZnP-Pz-PEO-COF was calculated to be 18.8 h$^{-1}$ (Fig. 5f). In contrast, the TOF of hydrophobic [Mo$_3$S$_{13}$]$^{2-}$@ZnP-Pz-COF, [Mo$_3$S$_{13}$]$^{2-}$@ZnP-Pz-DHTP-COF, amorphous [Mo$_3$S$_{13}$]$^{2-}$@ZnP-Pz-PEO-POP and [Mo$_3$S$_{13}$]$^{2-}$/ZnP-TP-PEO-COF was 0.016, 8.3, 3.8 and 1.7 h$^{-1}$, respectively (Fig. 5f). Therefore, the TOF of hydrophilic [Mo$_3$S$_{13}$]$^{2-}$@ZnP-Pz-PEO-COF is increased by more than 1200-fold as high as that of hydrophobic [Mo$_3$S$_{13}$]$^{2-}$@ZnP-Pz-COF. We investigated photocurrent generation by coating these photocatalysts on indium-tin oxide (ITO) substrates. Figure 5g shows the chopped photocurrent-versus-time plots. Clearly, [Mo$_3$S$_{13}$]$^{2-}$@ZnP-Pz-PEO-COF exhibited the highest photocurrent among the series, indicating that the best interface for electron flow from the light-harvesting antennae to the [Mo$_3$S$_{13}$]$^{2-}$ reaction centres is established.

To quantify the spectral contribution to the $H_2$ evolution, we evaluated the apparent quantum yield of [Mo$_3$S$_{13}$]$^{2-}$@ZnP-Pz-PEO-COF using band-pass filters. The quantum yield was determined to be 3.8%, 5.7%, 3.6% and 0.3% at 420, 500, 600 and 700 nm, respectively (Fig. 5h). This result indicates that the framework photocatalyst enables the use of a wide range of light for efficient $H_2$ production.

We investigated the performance stability of [Mo$_3$S$_{13}$]$^{2-}$@ZnP-Pz-PEO-COF. The [Mo$_3$S$_{13}$]$^{2-}$@ZnP-Pz-PEO-COF maintained a high photocatalytic activity after 18 h of six-cycle runs (Fig. 5i) and 12 h of continuous runs (Supplementary Fig. 18). Surprisingly, the PXRD results (Supplementary Fig. 19a) revealed that [Mo$_3$S$_{13}$]$^{2-}$@ZnP-Pz-PEO-COF retained its crystallinity after extended photocatalytic tests. Moreover,

the retained integrity of chemical structures of [Mo$_3$S$_{13}$]$^{2-}$@ZnP-Pz-PEO-COF was revealed by FTIR (Supplementary Fig. 19b). Element mapping with EDX of [Mo$_3$S$_{13}$]$^{2-}$@ZnP-Pz-PEO-COF (Supplementary Fig. 19c) demonstrated that Mo and S were still uniformly dispersed in the framework, which further supports that [Mo$_3$S$_{13}$]$^{2-}$@ZnP-Pz-PEO-COF is a stable photocatalyst.

## Exciton binding energy

To reveal insights on the separation and transport nature of exciton and charge carriers, we carried out temperature-dependent photoluminescence spectroscopy upon excitation at 580 nm. The luminescence intensity of crystalline ZnP-Pz-PEO-COF decreased gradually as the temperature was raised from 77 to 298 K (Fig. 6a), owing to the progress of thermally activated nonradiative recombination process[32]. The exciton binding energy of ZnP-Pz-PEO-COF was calculated to be 82 meV (Fig. 6b), which is higher than the thermal ionisation energy (26 meV), indicating that the transfer of photogenerated electron-hole pairs is favourable to be excitons rather than to be free electrons and holes[32]. The temperature-dependent fluorescence spectra of amorphous ZnP-Pz-PEO-POP (Fig. 6c) presented similar tendency to ZnP-Pz-PEO-COF. The exciton binding energy of ZnP-Pz-PEO-POP was 92 meV (Fig. 6d). The lower binding energy of ZnP-Pz-PEO-COF demonstrated that the excitons of ZnP-Pz-PEO-COF are easier to dissociate than those of ZnP-Pz-PEO-POP owing to the advantage of ordered π skeletons of crystalline ZnP-Pz-PEO-COF. These results are in accordance with the better photocatalytic activity of [Mo$_3$S$_{13}$]$^{2-}$@ZnP-Pz-PEO-COF.

## Femtosecond transient absorption and electron dynamics

We conducted femtosecond transient absorption (fs-TA) spectroscopy to reveal the photogenerated electron dynamics involved in the photocatalysis on the picosecond timescale[33–35]. The fs-TA spectra of ZnP-Pz-PEO-COF presents a broad negative signal from 420 to 550 nm (Fig. 6e), which can be assigned to the band-edge ground state bleach (GSB)[35]. Fig. 6f shows the dramatical change of the TA spectra with a new positive peak around 480 nm when upon adding the [Mo$_3$S$_{13}$]$^{2-}$ to the ZnP-Pz-PEO-COF. This new positive excited-state absorption band might originate from the reductive intermediate by the photoinduced electron compared to the fs-TA results of ZnP-Pz-PEO-COF (Fig. 6e) and (NH$_4$)$_2$Mo$_3$S$_{13}$ (Supplementary Fig. 20). Moreover, the UV–Vis spectrum of (NH$_4$)$_2$Mo$_3$S$_{13}$ in $CH_3CN$ (0.1 M $t$-Bu$_4$NPF$_6$) presented an enhancement of the absorption from 400 nm to 500 nm after 1 h electrolysis at −0.75 V under Ar, further suggesting that the new TA peak is from the reductive intermediate (Supplementary Fig. 21). Therefore, the kinetics of ZnP-Pz-PEO-COF probed at 500 nm were calculated to show the average lifetime ($\tau$) and rate constant ($k$) as 24 ps and 0.41 ns$^{-1}$, respectively (Fig. 6h, blue curve, Supplementary Table 7). Impressively, the [Mo$_3$S$_{13}$]$^{2-}$@ZnP-Pz-PEO-COF (Fig. 6f) exhibited a far shorter average lifetime (3 ps) and smaller rate constant (0.033 ns$^{-1}$) after loading [Mo$_3$S$_{13}$]$^{2-}$ clusters on the pore wall of ZnP-Pz-PEO-COF (Fig. 6h, red curve). The shortened lifetime indicated that the immobilisation of [Mo$_3$S$_{13}$]$^{2-}$ clusters created an additional pathway for the ultrafast photogenerated electron transfer from ZnP-Pz-PEO-COF to [Mo$_3$S$_{13}$]$^{2-}$ clusters[33–35]. This ultrafast electron transfer processes were also observed on ZnP-Pz-COF and ZnP-Pz-PEO-POP. The average lifetime of ZnP-Pz-COF, [Mo$_3$S$_{13}$]$^{2-}$@ZnP-Pz-COF, ZnP-Pz-PEO-POP and [Mo$_3$S$_{13}$]$^{2-}$@ZnP-Pz-PEO-POP was calculated to be 70, 35, 35 and 11.3 ps, respectively (Fig. 6g, i, Supplementary Fig. 22). Moreover, the average lifetime of crystalline hydrophilic [Mo$_3$S$_{13}$]$^{2-}$@ZnP-Pz-PEO-COF is shorter than those of crystalline hydrophobic [Mo$_3$S$_{13}$]$^{2-}$@ZnP-Pz-COF and amorphous hydrophilic [Mo$_3$S$_{13}$]$^{2-}$@ZnP-Pz-PEO-POP, which are consistent with the highest $H_2$ evolution rate of [Mo$_3$S$_{13}$]$^{2-}$@ZnP-Pz-PEO-COF among the series. These new insights demonstrate that the integrated interfacial design of π-electronic interface and hydrophilic nanopores promote the photocatalytic reaction.

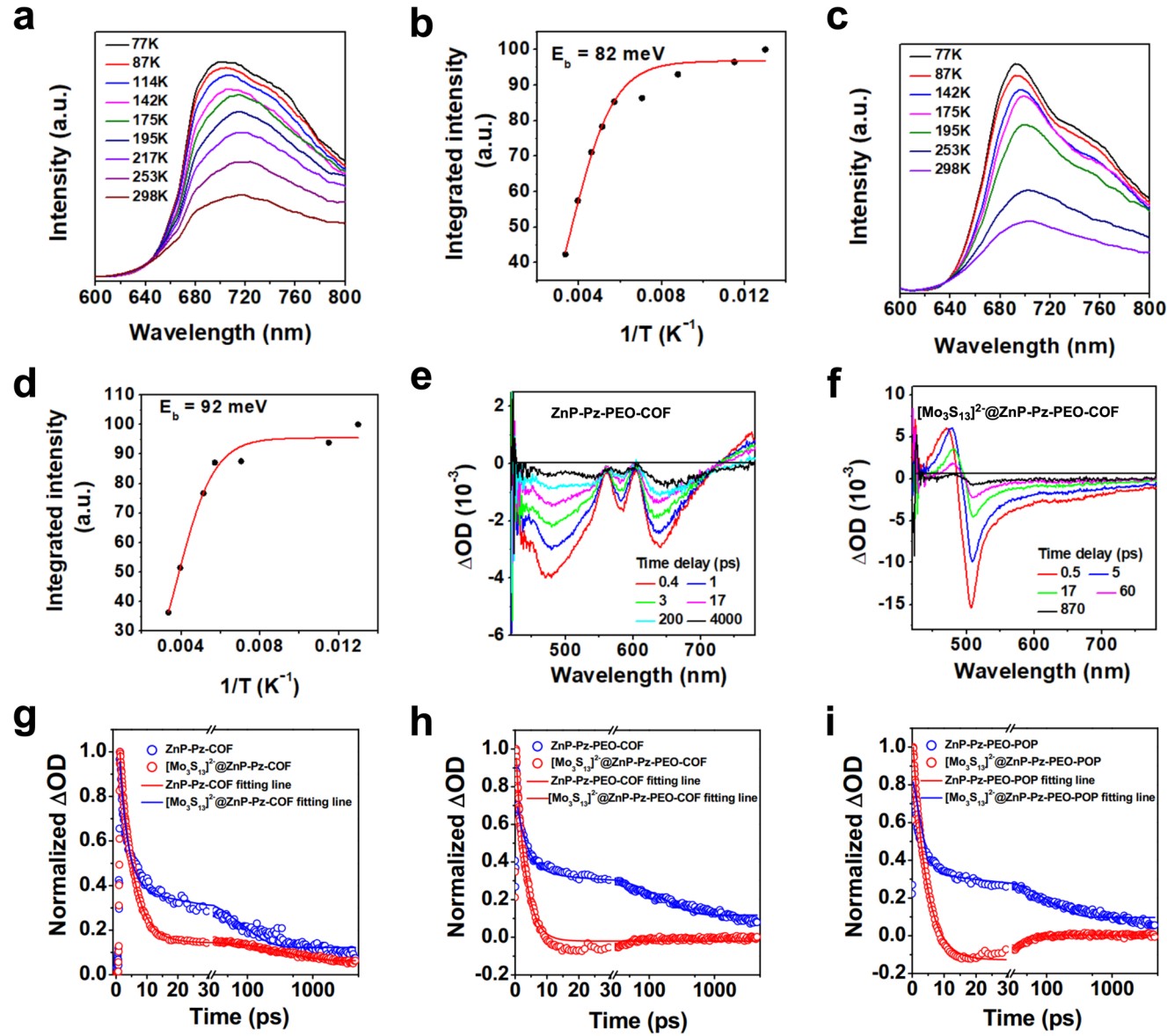

**Fig. 6 | Photophysical property. a** Temperature-dependent photoluminescence spectra of ZnP-Pz-PEO-COF with excitation wavelength at 580 nm. **b** Extracted exciton binding energy of ZnP-Pz-PEO-COF. **c** Temperature-dependent photoluminescence spectra of ZnP-Pz-PEO-POP with excitation wavelength at 586 nm. **d** Extracted exciton binding energy of ZnP-Pz-PEO-POP. Femtosecond transient absorption spectra of **e** ZnP-Pz-PEO-COF and **f** [Mo$_3$S$_{13}$]$^{2-}$@ZnP-Pz-PEO-COF, pumped at 400 nm. **g** Comparison of the kinetics for ZnP-Pz-COF and [Mo$_3$S$_{13}$]$^{2-}$@ZnP-Pz-COF probed at 500 nm. **h** Comparison of the kinetics for ZnP-Pz-PEO-COF and [Mo$_3$S$_{13}$]$^{2-}$@ZnP-Pz-PEO-COF probed at 500 nm. **i** Comparison of the kinetics for ZnP-Pz-PEO-POP and [Mo$_3$S$_{13}$]$^{2-}$@ZnP-Pz-PEO-POP probed at 500 nm.

## Discussion

Our studies on elucidating the key fundamental interfaces for photoinduced hydrogen evolution from water with non-noble metal catalytic centres unambiguously reveal the necessary structures for photocatalysts and lead to the finding of the best photocatalysts. By setting the structural parameters on skeleton for electron flow, coordination sites for ligating metal centre and nanochannels for mass transport, we predesigned the interfaces for each process and integrated them into the framework materials. Remarkably, the change of structures for these key processes leads to a profound effect on the photocatalytic reaction. Changing the skeleton from amorphous polymer to crystalline framework greatly improves the photocatalytic activity by facilitating electron flow. Changing non-ligating walls to nitrogen-rich ligating pore walls improves the loading of metal centres onto the proximate location on the skeleton, so that photogenerated electrons can be quickly transported from the skeleton to the reaction centre,

greatly improving the photocatalytic efficiency. A systematic changing of nanochannels from hydrophobic to hydrophilic enables to construct a wide structural spectrum to facilitate the water delivery to the reaction centre and greatly promotes the photocatalytic reaction. Through comparative studies on counterpart systems with amorphous, hydrophobic and non-ligating walls, the role of each process involved in the photocatalytic reaction becomes clear. The COFs, combined crystalline skeleton, hydrophilic channels and ligating walls, can merge electron transfer, electron flow and mass transport into the photocatalytic cycle to achieve the highest performance. In this sense, this interfacial design approach is a step towards designing photocatalysts via bottom-up structural control.

In summary, we have developed an integrated interfacial design strategy for constructing active and robust photocatalysts for hydrogen production from water. We designed the photocatalyst by engineering three distinct interfaces to facilitate electron flow and water

transport to the reaction centre: (1) For the π-electronic interface, the π skeleton was designed to efficiently harvest photons extended to 1000 nm and build ordered π skeletons to promote a seamless electron transfer from the antennae to the catalytic centres; (2) for the non-noble metal immobilisation interface, the walls were installed with desired ligating sites; and (3) for the mass transport interface, the nanopores were engineered to be hydrophilic to facilitate water delivery to the reaction centres. Remarkably, the resultant photocatalysts work with a non-noble metal co-catalyst and enable the use of a wide range of photons to achieve high evolution rate, quantum yield and turnover frequency, which are far superior to those obtained with the state-of-the-art noble metal platinum systems. These results disclosed unprecedented insights that integrated interfacial design is key to photocatalytic hydrogen evolution. We envision that our approach is widely applicable to other photocatalytic systems and open the way to actionable solar-to-chemical energy conversion and green fuel production.

## Methods

### ZnP-Pz-COF

An o-DCB/dioxane (1/1 in vol, 1 mL) mixture of ZnP (14.6 mg, 0.02 mmol) and pyrazine-2,5-dialdehyde (5.4 mg, 0.04 mmol) in the presence of acetic acid (6 M, 0.1 mL) was degassed in a Pyrex tube (10 mL) by three freeze-pump-thaw cycles. The tube was sealed off and heated at 120 °C for 3 days. The precipitate was collected by filtration, washed with THF and subjected to Soxhlet extraction with THF for 1 day. The powder was collected and dried at room temperature under vacuum overnight to give ZnP-Pz-COF in an isolated yield of 88%.

### ZnP-Pz-DHTP-COF

An o-DCB/dioxane (1/1 in vol, 1 mL) mixture of ZnP (14.6 mg, 0.02 mmol), pyrazine-2,5-dialdehyde (2.7 mg, 0.02 mmol) and 2,5-dihydroxyterephthalaldehyde (3.3 mg, 0.02 mmol) in the presence of acetic acid (6 M, 0.1 mL) was degassed in a Pyrex tube (10 mL) by three freeze-pump-thaw cycles. The tube was sealed off and heated at 120 °C for 3 days. The precipitate was collected by filtration, washed with THF and subjected to Soxhlet extraction with THF for 1 day. The powder was collected and dried at room temperature under vacuum overnight to give ZnP-Pz-DHTP-COF in an isolated yield of 86%.

### ZnP-Pz-DHTP-POP

An o-DCB (1 mL) of ZnP (14.6 mg, 0.02 mmol), pyrazine-2,5-dialdehyde (2.7 mg, 0.02 mmol), and 2,5-dihydroxyterephthalaldehyde (3.3 mg, 0.02 mmol) in the presence of acetic acid (6 M, 0.1 mL) was degassed in a Pyrex tube (10 mL) by three freeze-pump-thaw cycles. The tube was sealed off and heated at 120 °C for 3 days. The precipitate was collected by filtration, washed with THF and subjected to Soxhlet extraction with THF for 1 day. The powder was collected and dried at room temperature under vacuum overnight to give ZnP-Pz-DHT-POP in an isolated yield of 81%.

### ZnP-TP-DHTP-COF

An o-DCB/n-butanol (1/1 in vol, 1 mL) mixture of ZnP (14.6 mg, 0.02 mmol), terephthalaldehyde (2.7 mg, 0.02 mmol) and 2,5-dihydroxyterephthalaldehyde (3.3 mg, 0.02 mmol) in the presence of acetic acid (6 M, 0.1 mL) was degassed in a Pyrex tube (10 mL) by three freeze-pump-thaw cycles. The tube was sealed off and heated at 120 °C for 3 days. The precipitate was collected by filtration, washed with THF and subjected to Soxhlet extraction with THF for 1 day. The powder was collected and dried at room temperature under vacuum overnight to give ZnP-TP-DHTP-COF in an isolated yield of 85%.

### ZnP-Pz-PEO-COF, ZnP-Pz-PEO-POP and ZnP-TP-PEO-COF

A ZnP-Pz-DHTP-COF sample (20 mg, 0.04 mmol −OH) was dispersed in DMF (4 mL) and sonicated for 15 min. A mixture of 1-bromo-2-(2-methoxyethoxy)ethane (100 μL, 0.8 mmol) and $K_2CO_3$ (110 mg, 0.8 mmol) was added to the above solution. The resulting mixture was stirred at 85 °C for 24 h. After the mixture was cooled to room temperature, the crude product was collected by filtration, washed with water, acetone, and THF and subjected to Soxhlet extraction with THF for 12 h. The powder was collected and dried at room temperature under vacuum overnight to give ZnP-Pz-PEO-COF in an isolated yield of 96% (based on the 93% of PEO and 7% of −OH units which were determined by elemental analysis). Similarly, ZnP-Pz-DHTP-COF was replaced with ZnP-Pz-DHTP-POP and ZnP-TP-DHTP-COF to give ZnP-Pz-PEO-POP and ZnP-TP-PEO-COF in a yield of 98% and 96%, respectively.

### $[Mo_3S_{13}]^{2-}$@ZnP-Pz-COF

A $(NH_4)_2Mo_3S_{13}$ sample (29 mg, 0.040 mmol) in methanol (10 mL) was sonicated for 30 min and added with ZnP-Pz-COF (50 mg, 0.106 mmol pyrazine unit). The mixture was sonicated for 30 min and stirred at 75 °C for 24 h. After cooled to room temperature, the precipitate was collected by centrifugation at 5724×g for 5 min and washed with methanol for three times. The $[Mo_3S_{13}]^{2-}$@ZnP-Pz-COF was collected and dried at room temperature under vacuum overnight. The content of Mo was 6.21%.

### $[Mo_3S_{13}]^{2-}$@ZnP-Pz-DHTP-COF

A $(NH_4)_2Mo_3S_{13}$ sample (38 mg, 0.052 mmol) in methanol (10 mL) was sonicated for 30 min and added with ZnP-Pz-DHTP-COF (50 mg, 0.052 mmol pyrazine unit). The mixture was sonicated for 30 min and stirred at 75 °C for 24 h. After cooled to room temperature, the precipitate was collected by centrifugation at 5724×g for 5 min and washed with methanol for three times. The $[Mo_3S_{13}]^{2-}$@ZnP-Pz-DHTP-COF was collected and dried at room temperature under vacuum overnight. The content of Mo was 5.55%.

### $[Mo_3S_{13}]^{2-}$@ZnP-Pz-PEO-COF, $[Mo_3S_{13}]^{2-}$@ZnP-Pz-PEO-POP, and $[Mo_3S_{13}]^{2-}$/ZnP-TP-PEO-COF

A $(NH_4)_2Mo_3S_{13}$ sample (32 mg, 0.043 mmol) in methanol (10 mL) was sonicated for 30 min and added with ZnP-Pz-PEO-COF (50 mg, 0.043 mmol pyrazine unit). The mixture was sonicated for 30 min and stirred at 75 °C for 24 h. After cooled to room temperature, the precipitate was collected by centrifugation at 5724×g for 5 min and washed with methanol for three times. The $[Mo_3S_{13}]^{2-}$@ZnP-Pz-PEO-COF was collected and dried at room temperature under vacuum overnight. The content of Mo was measured to be 5.61%. A $(NH_4)_2Mo_3S_{13}$ sample of 16 mg and 64 mg was used to yield $[Mo_3S_{13}]^{2-}$@ZnP-Pz-PEO-COF with the Mo content of 3.22% and 8.27%, respectively. ZnP-Pz-PEO-COF was replaced with ZnP-Pz-PEO-POP (50 mg) and ZnP-TP-PEO-COF (50 mg) to yield $[Mo_3S_{13}]^{2-}$@ZnP-Pz-PEO-COF and $[Mo_3S_{13}]^{2-}$/ZnP-TP-PEO-COF with the Mo content of 5.48% and 4.94%, respectively.

### Photocatalytic hydrogen evolution

Typically, the photocatalyst (10 mg) was dispersed in an aqueous solution (10 mL) with sacrificial donor (ascorbic acid or lactic acid). The suspension was then bubbled with nitrogen to remove the residual oxygen before sealed in a quartz flask. The photocatalytic hydrogen evolution was carried out by irradiating the mixture with a 300-W Xenon lamp (MAX-303, Asahi Spectra, Japan) and a 420 nm cutoff filter. The amount of hydrogen was measured by using gas chromatograph (Aglient 7890 A, TCD, 13 X columns, S-6 Ar carrier).

## Data availability

All data are available in the main text or the supplementary information.

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

## Acknowledgements

This work is supported from Singapore MOE Tier 2 grant (MOE-T2EP10220-0004 and MOE-T2EP10221-0006), MOE Tier1 grants (A-0008368-00-00 and A-0008369-00-00) and A*star LCER-F1 project (U2102d2004). W.C. acknowledges the support of NUS Flagship Green Energy Program. W.Z. and C.X. thank the support from Singapore MOE Tier 2 grant (MOE2018-T2-1-017) and MOE Tier1 grants (MOE2019-T1-002-012, RG102/19). Y. Y.C. acknowledges the financial support from the China Scholarship Council (201906150104).

## Author contributions

D.J. conceived the project, designed experiments and provided funds. T.H., W.Z., Zhu.L., Y.C., Zho.L., R.L. and C.X. conducted the experiments. N.H. performed the calculations. Y.L., X.L. and W.C. conducted the XPS experiments. Y.G. and T.C.S. and performed transient absorption spectroscopy. D.J. and T.H. wrote the manuscript and discussed the results with all authors. All data are reported in the main text and supplementary materials.

## Competing interests
The authors declare no competing interests.
