## [Peer Review File · Nature Communications]

Integrated interfacial design of covalent organic framework photocatalysts to promote hydrogen evolution from waterReviewers' comments:

Reviewer #1 (Remarks to the Author):

The manuscript by Jiang and co-workers describes the preparation and characterization of novel COF materials embodying ZnP chromophores, a molybdenum based catalyst and specific functional groups to introduce hydrophilicity or hydrophobicity in the material's pores and employed them for light-driven hydrogen evolution in aqueous solution. The results are interesting and will be of significance in the field. The methodology employed as well as the results reported are sound and complete, particularly as far as the characterization of the novel materials is concerned. I may recommend publication of the present manuscript after the authors have addressed the following comments.

- p.2 line 54. What do the authors mean with wide-range visible lights? Please clarify.
- fig. 1b. The molecular structures are not clearly visible. Please improve the resolution of the image for clarity.
- p. 8, line 225. The authors report the comparison between ascorbic acid and lactic acid and infer on the superior activity by lactic acid. However, the concentrations of the two donors are different so that the comparison is not strictly pertinent. Furthermore, the authors investigated the role of the donor (either ascorbic acid or lactic acid) without considering the resulting pH. The oxidation of both ascorbic and lactic acid is indeed expected to be pH dependent so that the activity of both compounds as donors might be different as a function of both the concentration and pH. I would suggest to control the effect of the donor by fixing the pH in order to disentangle the mutual effect of pH and donor concentration.
- connected to the previous point. Evaluation of the pH effect (at the same donor concentration) should be also considered in the optimization of the photocatalytic performance.
- p. 9, line 267. The authors used TOF as a metric to compare the photocatalytic activity but did not describe in detail what they refer to with TOF. Please clarify.
- p. 10, line 274. There is no discussion after that line. Please remove.
- p. 10, line 285. The authors investigated the stability of the most performing sample and refer to 6 cycle photocatalysis. Did the authors add fresh electron donor after each run? Did they try a direct 12 h photocatalysis?
- p. 10, line 280. The photocurrent measurements are in my opinion superfluous since these data should be implemented with faradaic yield measurements and JV curves.
- p. S3. As to the estimate of the AQY, how did the authors calculate the intensity of the irradiation? Why did the authors choose 1 h time for estimating the AQY? This quantity must be estimated under conditions where the production of hydrogen is linear vs. time and the authors should check whether this conditions is met herein.

Reviewer #2 (Remarks to the Author):

Jiang et al. describe a case-study that merges order, ligation, and hydrophilicity features into one framework photocatalyst for hydrogen evolution. By comparing the performances of hydrogen evolution from a family of porous materials, the authors conclude all the three features collectively contribute the impressive hydrogen evolution yield of [Mo₃S₁₃]²⁻@ZnP-Pz-PEO-COF. This seems to be a superficial conclusion without significant scientific merits. Given the novelty of the work, I cannot recommend the acceptance of this manuscript for the publication in Nature Communications.

Other following issues that I observed from the submitted documents also raise some concerns of accepting the current form of the manuscript:

- The manuscript title would be better presented by a phrase, rather than a sentence.
- The referencing in the manuscript is outdated or not responsible. For example, these critically related references were not included (Polym. Chem., 2021, 12, 3250-3256; Nature Communications, 2022, 13, 2357; ACS Energy Lett., 2018, 3, 400).
- In the section of Crystalline Structures on Page 4, several PXRD patterns present a peak of $\sim 4.2^\circ$, which is totally ignored and not discussed by the authors.
- The proposed COF structure in Figure 1, obtained by reacting Zn-based tetrakis(4-aminophenyl)porphyrin (ZnP) and two types of aldehyde molecules (Pz and DHTP) in one pot, illustrates Pz molecules being aligned in the para position. The same is highlighted to the DHTP molecules. Do authors have any experimental evidence to prove this kind of spatial arrangement?
- As for the Mo-included COFs, the Mo-content is reported as 4.9 – 6.2 wt% using ICP-OES. It would be better to present the data by showing the ratios of Mo/Zn, which can highlight the number of pyrazine sites being occupied by Mo₃ cluster.
- Could the authors provide the error bars for the yields of photocatalysis reactions?
- I am curious about how the turnover frequency (TOF) being calculated for the heterogeneous catalysts. The concept of TON or TOF is usually applied to homogeneous catalysts, as we really don't know the numbers of active sites participating in the reaction (the diffusion issue). In this case, the light penetration could be a problem, as the inner sites may not be involved.

Reviewer #1 (Remarks to the Author):

The manuscript by Jiang and co-workers describes the preparation and characterization of novel COF materials embodying ZnP chromophores, a molybdenum based catalyst and specific functional groups to introduce hydrophilicity or hydrophobicity in the material's pores and employed them for light-driven hydrogen evolution in aqueous solution. The results are interesting and will be of significance in the field. The methodology employed as well as the results reported are sound and complete, particularly as far as the characterization of the novel materials is concerned. I may recommend publication of the present manuscript after the authors have addressed the following comments.

1. p.2 line 54. What do the authors mean with wide-range visible lights? Please clarify.

Answer: We appreciate your kind suggestions.

We have revised the "wide-range visible lights" to "wide-range visible light absorption (from 300 to over 1,000 nm).".

2. fig. 1b. The molecular structures are not clearly visible. Please improve the resolution of the image for clarity.

Answer: We appreciate your kind suggestions.

We have improved the resolution of fig. 1b for a better reading.

3. p. 8, line 225. The authors report the comparison between ascorbic acid and lactic acid and infer on the superior activity by lactic acid. However, the concentrations of the two donors are different so that the comparison is not strictly pertinent. Furthermore, the authors investigated the role of the donor (either ascorbic acid or lactic acid) without considering the resulting pH. The oxidation of both ascorbic and lactic acid is indeed expected to be pH dependent so that the activity of both compounds as donors might be different as a function of both the concentration and pH. I would suggest to control the effect of the donor by fixing the pH in order to disentangle the mutual effect of pH and donor concentration.

Answer: We appreciate your kind suggestions. We used ascorbic acid (1.17 M; pH = 1.46) as sacrificial donor with the same concentration and pH value of 15 % (V/V) lactic acid. The H₂ evolution rate was quite lower (1.41 mmol g⁻¹ h⁻¹) than that of lactic acid conditions (Supplementary Figs. 13 and 14). We have added the part "To exclude the effect of pH values and concentrations between ascorbic acid and lactic acid, we adjusted the pH value of 1.17 M ascorbic acid to 1.46 with diluted HCl aqueous, which presented the same concentration and pH value of 15 % (V/V) lactic acid. The H₂ evolution rate was only 1.41 mmol g⁻¹ h⁻¹ (Supplementary Figs. 13 and 14), revealing the better performance of lactic acid.", for a better understanding.

4. Connected to the previous point. Evaluation of the pH effect (at the same donor concentration) should be also considered in the optimization of the photocatalytic

performance.

Answer: We appreciate your kind suggestions. We have evaluated the pH effect by fixing the concentration of lactic acid as 15 % (V/V). The lactic acid conditions with pH value as 4 showed the highest H₂ evolution rate as 14.5 mmol g⁻¹ h⁻¹ (Supplementary Figs. 13 and 14). Increasing or decreasing the pH value of lactic acid aqueous resulted in the lower H₂ evolution rate. We have added the part “Interestingly, the H₂ production rate demonstrated a volcano-like trend related to the pH value of lactic acid aqueous, with a highest rate of 14.5 mmol g⁻¹ h⁻¹ (Supplementary Figs. 13 and 14) over 15 % (V/V) lactic acid with pH value of 4.”, for a better understanding.

5. p. 9, line 267. The authors used TOF as a metric to compare the photocatalytic activity but did not describe in detail what they refer to with TOF. Please clarify.

Answer: We appreciate your kind suggestions. We have revised the part “We used turnover frequency (TOF) based on the amount of Mo to evaluate the photocatalytic activity.” in the manuscript. Moreover, We added the calculation details of TOF in Supporting Information.

“Turnover frequency (TOF)

TOF is a kinetic-dependent parameter, which was calculated according to the following equation:

$$TOF = \frac{\text{Amount of H}_2 \text{ molecules evolved in unit time (mmol g}^{-1} \text{ h}^{-1})}{\text{Amount of Mo (mmol g}^{-1})} \text{ , ,}$$

6. p. 10, line 274. There is no discussion after that line. Please remove.

Answer: We appreciate your kind suggestions. We have removed this part to “Results” part and added a new “Discussion” part in the manuscript based on the temperature-dependent photoluminescence results and femtosecond transient absorption results.

7. p. 10, line 285. The authors investigated the stability of the most performing sample and refer to 6 cycle photocatalysis. Did the authors add fresh electron donor after each run? Did they try a direct 12 h photocatalysis?

Answer: We appreciate your kind suggestions. We supplied 0.2 mL lactic acid after the third cycle. We also conducted the stability experiment for a direct 12 h photocatalysis. The H₂ evolution rate maintained after direct 12 h photocatalysis (Supplementary Fig. 17). We revised the part “The [Mo₃S₁₃]²⁻@ZnP-Pz-PEO-COF maintained a high photocatalytic activity after 18 h of six-cycle runs (Fig. 5i) and 12 h of continuous runs (Supplementary Fig. 17).” for a better understanding.

8. p. 10, line 280. The photocurrent measurements are in my opinion superfluous since these data should be implemented with faradaic yield measurements and JV curves.

Answer: We appreciate your kind suggestions. The photocurrent measurements are

always conducted to evaluate the excitation of the photo-induced electron–hole pairs and characterize carrier separation during the photocatalysis process (*Angew. Chem. Int. Ed.*, **2022**, *61*, e2022043. *Nat. Commun.*, **2021**, *12*, 1354. *Adv. Mater.*, **2020**, *32*, 1906361. *Nat. Commun.*, **2018**, *9*, 3366.). The faradaic yield measurements and JV curves are always measured to evaluate the activity of catalysts involving electrocatalysis. Therefore, we think that the photocurrent results here are necessary to support our conclusion.

9. p. S3. As to the estimate of the AQY, how did the authors calculate the intensity of the irradiation? Why did the authors choose 1 h time for estimating the AQY? This quantity must be estimated under conditions where the production of hydrogen is linear vs. time and the authors should check whether this conditions is met herein.

Answer: We appreciate your kind suggestions. The intensity of irradiation light was measured by a solar power meter. We have also recorded the time-dependent H₂ production profiles monitored over 3 h with different bandpass filters of 420, 500, 600, and 700 nm and used the average amount of H₂ of one hour to calculate AQE. We have revised the part “AQE was measured under the same conditions by using the same Xenon lamp equipped with different bandpass filters of 420, 500, 600, and 700 nm. The intensity of irradiation light was measured by solar power meter (Newport, Model 1918-R). The average amount of H₂ was measured to calculate AQE in according to the following equation.” in supporting information for a better understanding.

We appreciate this reviewer for suggestive comments to improve the quality.

Reviewer #2 (Remarks to the Author):

Jiang et al. describe a case-study that merges order, ligation, and hydrophilicity features into one framework photocatalyst for hydrogen evolution. By comparing the performances of hydrogen evolution from a family of porous materials, the authors conclude all the three features collectively contribute the impressive hydrogen evolution yield of $[\text{Mo}_3\text{S}_{13}]^{2-}@ZnP\text{-Pz-PEO-COF}$. This seems to be a superficial conclusion without significant scientific merits. Given the novelty of the work, I cannot recommend the acceptance of this manuscript for the publication in Nature Communications.

Answer: We appreciate your kind suggestions.

In this work, we report a strategy for integrated interfacial designs to differentiate molecular interfaces to control electron transfer, active centre immobilisation, and water transport, with an aim to define highly active and robust photocatalysts to drive H_2 production from water. The $[\text{Mo}_3\text{S}_{13}]^{2-}@ZnP\text{-Pz-PEO-COF}$ with ordered π skeletons, ligating walls, and hydrophilic channels exhibited the highest H_2 evolution rate over $11 \text{ mmol g}^{-1} \text{ h}^{-1}$, a quantum yield of 3.6% at 600 nm. The H_2 evolution rate is much far superior which is two orders of magnitude as high as those of reported non-noble metal-based COF systems.

To investigate deep insights about the separation and transfer of exciton and charge carriers in $[\text{Mo}_3\text{S}_{13}]^{2-}@ZnP\text{-Pz-PEO-COF}$ during the photocatalysis, we measured temperature-dependent photoluminescence (PL) and femtosecond transient absorption (fs-TA) spectroscopy.

The exciton binding energy of $ZnP\text{-Pz-PEO-COF}$ and $ZnP\text{-Pz-PEO-POP}$ is calculated to be 82 and 92 meV, respectively (Figs. 6b and d) through PL results, which are higher than thermal ionization energy ($\sim 26 \text{ meV}$), indicating the transfer of photogenerated electron-hole pairs are favourable to be excitons rather than to be free electrons and holes. Moreover, the lower binding energy of $ZnP\text{-Pz-PEO-COF}$ demonstrates that the excitons of $ZnP\text{-Pz-PEO-COF}$ are more prone to dissociation than those of $ZnP\text{-Pz-PEO-POP}$ owing to the advantage of crystalline π skeletons of $ZnP\text{-Pz-PEO-COF}$.

The average life time (τ) of $ZnP\text{-Pz-PEO-COF}$ is calculated to be 24 ps through fs-TA (Figs. 6e and h). Impressively, the $[\text{Mo}_3\text{S}_{13}]^{2-}@ZnP\text{-Pz-PEO-COF}$ shows much shorter average lifetime of 3 ps after loading $[\text{Mo}_3\text{S}_{13}]^{2-}$ clusters on the pore wall of $ZnP\text{-Pz-PEO-COF}$ (Figs. 6f and h). The shorten lifetime indicates that the immobilisation of $[\text{Mo}_3\text{S}_{13}]^{2-}$ clusters creates an additional pathway for the ultrafast photogenerated electron transfer from $ZnP\text{-Pz-PEO-COF}$ to $[\text{Mo}_3\text{S}_{13}]^{2-}$ clusters. Furthermore, the average lifetime of crystalline and hydrophilic $[\text{Mo}_3\text{S}_{13}]^{2-}@ZnP\text{-Pz-PEO-COF}$ was shorter than those of crystalline and hydrophobic $[\text{Mo}_3\text{S}_{13}]^{2-}ZnP\text{-Pz-COF}$ (35 ps) and amorphous and hydrophilic $[\text{Mo}_3\text{S}_{13}]^{2-}ZnP\text{-Pz-PEO-POP}$ (11.3 ps), demonstrating the integrated interfacial design of π -electronic interface and hydrophilic nanopores to promote the electron transfer involving the photocatalysis, thus leading to the highest H_2 evolution rate of

[Mo₃S₁₃]²⁻@ZnP-Pz-PEO-COF.

We firmly believe that the excellent photocatalytic H₂ evolution performance as well as the deep investigation of the photophysical process provides a breakthrough strategy for the development of non-noble metal-based polymeric photocatalytic systems for the water reduction.

Other following issues that I observed from the submitted documents also raise some concerns of accepting the current form of the manuscript:

1. The manuscript title would be better presented by a phrase, rather than a sentence.

Answer: We appreciate your kind suggestions.

We have revised the title to be “Integrated interfacial design of covalent organic framework photocatalysts to promote hydrogen evolution from water”.

2. The referencing in the manuscript is outdated or not responsible. For example, these critically related references were not included (Polym. Chem., 2021, 12, 3250-3256; Nature Communications, 2022, 13, 2357; ACS Energy Lett., 2018, 3, 400).

Answer: We appreciate your kind suggestions. We have updated the recent references about COF photocatalysts for H₂ evolution. Those are:

“Ref. 18. Li, C. et al. Covalent organic frameworks with high quantum efficiency in sacrificial photocatalytic hydrogen evolution. *Nat. Commun.* **13**, 2357 (2022).

Ref. 19. Li, Y. et al. In situ photodeposition of platinum clusters on a covalent organic framework for photocatalytic hydrogen production. *Nat. Commun.* **13**, 1355 (2022).

Ref. 20. Sun, L. et al. Nickel glyoximate based metal-covalent organic frameworks for efficient photocatalytic hydrogen evolution. *Angew. Chem. Int. Ed.* **61**, e202204326 (2022).

Ref. 21. Yang, S. et al. Transformation of covalent organic frameworks from N-acylhydrazone to oxadiazole linkages for smooth electron transfer in photocatalysis. *Angew. Chem. Int. Ed.* **61**, e202115655 (2022).

Ref. 22. Zhou, T. et al. Multivariate covalent organic frameworks boosting photocatalytic hydrogen evolution. *Polym. Chem.* **12**, 3250–3256 (2021).

Ref. 23. Banerjee, T., Gottschling, K., Savasci, G., Ochsenfeld, C. & Lotsch, B. V. H₂ evolution with covalent organic framework photocatalysts. *ACS Energy Lett.* **3**, 400–409 (2018).”

3. In the section of Crystalline Structures on Page 4, several PXRD patterns present a peak of ~4.2 °, which is totally ignored and not discussed by the authors.

Answer: We appreciate your kind suggestions. We have discussed the peak at 4 degree and claimed it as “The small peak at 4.0° might be attributed to the formation of some extended slipped J-aggregate stacks, which was observed in other porphyrin-based

COFs³⁰.”

Ref. 30. Keller, N. et al. Enforcing extended porphyrin *J*-aggregate stacking in covalent organic frameworks. *J. Am. Chem. Soc.* **140**, 16544–16552 (2018).

4. The proposed COF structure in Figure 1, obtained by reacting Zn-based tetrakis(4-aminophenyl)porphyrin (ZnP) and two types of aldehyde molecules (Pz and DHTP) in one pot, illustrates Pz molecules being aligned in the para position. The same is highlighted to the DHTP molecules. Do authors have any experimental evidence to prove this kind of spatial arrangement?

Answer: We appreciate your kind suggestions. Actually, we have no experimental evidence to confirm the spatial arrangement of two kinds of aldehydes except we can achieve the single crystals of COFs. However, this is one of the most difficult topics in the field of 2D COFs. We can confirm that the ratio of Pz and DHTP is 1 to 1, so we arrange the two units on the para position of COF structures for a better understanding.

5. As for the Mo-included COFs, the Mo-content is reported as 4.9 – 6.2 wt% using ICP-OES. It would be better to present the data by showing the ratios of Mo/Zn, which can highlight the number of pyrazine sites being occupied by Mo₃ cluster.

Answer: We appreciate your kind suggestions. We have added the theoretical molar ratios of Zn and Mo when nitrogen atoms of pyrazine units are fully coordinated with [Mo₃S₁₃]²⁻ and determined molar ratios of Zn and Mo for all samples in Supplementary Table 4.

6. Could the authors provide the error bars for the yields of photocatalysis reactions?

Answer: We appreciate your kind suggestions. We have added error bars for the hydrogen evolution rates of different samples in fig. 5e.

7. I am curious about how the turnover frequency (TOF) being calculated for the heterogeneous catalysts. The concept of TON or TOF is usually applied to homogeneous catalysts, as we really don't know the numbers of active sites participating in the reaction (the diffusion issue). In this case, the light penetration could be a problem, as the inner sites may not be involved.

Answer: We appreciate your kind suggestions. The TON and TOF are normally used in homogeneous catalytic systems. The most important issue is how to precisely calculate the number of active sites when TOF is used in heterogeneous catalytic systems. The TOF in our case is calculated based on the amount of Mo in catalysts. We think the relative TOF values are suitable for the comparison of activity of different catalysts in the same experimental conditions if we assumed that all Mo

atoms participated in the photocatalysis. Moreover, the TOF are also used in other published papers for comparison. (*Energy Environ. Sci.*, **2015**, *8*, 2668–2676. *Adv. Mater.*, **2016**, *28*, 2427–2431. *Angew. Chem. Int. Ed.*, **2018**, *57*, 12106–12110. *Adv. Funct. Mater.*, **2018**, *28*, 1802169. *ACS Catal.*, **2021**, *11*, 13266–13279. *Nat. Commun.*, **2022**, *13*, 1287.

We appreciate this reviewer for suggestive comments to improve the quality.

REVIEWER COMMENTS

Reviewer #1 (Remarks to the Author):

All the concerns raised by the reviewers have been adequately addressed. The addition of ultrafast experiments is also a plus that is consistent with the photocatalytic results.

The manuscript can be publishable provided that the authors address the following last comments.

-) P. 13, line 335, and fig. 6f. The authors describe the substantial difference between the transient absorption of the COF and the COF+catalyst. However, no detailed explanation is given as to the featuring spectral signatures observed in the latter system (fig. 6f). These spectral features should be properly assigned in order to confirm the observed behavior and compare the corresponding decay kinetics.

-) The temperature dependent emission results and the ultrafast experiments on pp. 12,13 should be described on the "Results" section. A proper "Discussion" section should be added which discusses in a more comprehensive manner the results previously outlined.

Reviewer #2 (Remarks to the Author):

I appreciate the authors' efforts on addressing the concerns and questions that were raised earlier. There are still missing details of this research and I am not convinced that the current quality and novelty of the work warrant the acceptance of Nat. Commun.

1) Novelty issue. Multivariate COFs (Polym. Chem., 2021,12, 3250) and Mo-cluster encapsulated COFs (Chem. Commun., 2018, 54, 13563; ACS Energy Lett. 2018, 3, 400) have been both evaluated for hydrogen production as photocatalysts. The hydrophilic motif was also investigated from the previous ref (Angew. Chem. Int. Ed., 2019, 58, 18290). This work integrates all the three components in one system for photocatalytic H₂ production and is attractive to certain audience. It seems that a more specific journal would be an appropriate venue for the publication.

2) "The small peak at 4.0° might be attributed to the formation of some extended slipped J-aggregate stacks, which was observed in other porphyrin-based COFs.³⁰" This explanation is not satisfactory and I am not able to identify any 4.0° discussion in the cited reference.

3) "We can confirm that the ratio of Pz and DHTP is 1 to 1, so we arrange the two units on the para position of COF structures for a better understanding." Has the resultant 1:1 ratio between Pz and DHTP been experimentally confirmed, though the initial experimental loading was 1:1? I didn't this experimental evidence in neither manuscript nor SI. A better understanding does not come from the arbitrary assignment of chemical species in the space without any explanation. An explanation or illustration for this arbitrary para-arrangement has to be added to the figure caption.

4) The resolution of Figure 1 has not been improved, as pointed by the other referee.

5) Figure 3, EDX elemental mapping for Zn was not clearly illustrated, probably due to the demetallation during the synthesis. Given the stability of Zn-base porphyrin has relatively low tolerance toward protons, I am a little concerned by the demetallation of the porphyrin units during the synthesis (exposed to HOAc). Any experimental data to disprove the demetallation? Is the role of Zn critical for light harvesting purpose?

6) Page 4, line 90. "The hydrophilic ZnP-Pz-PEO-COF with pyrazine sites was synthesised in 96% yield..." How has this reaction yield been determined? Please be specific. This number is slightly different from data on Page 6, line 146, which only indicate 93% of -OH groups involve in the reaction.

7) Page 10, line 256, "Remarkably, [Mo₃S₁₃]@ZnP-Pz-PEO-COF is much superior to those of the state-of-the-art Pt-based systems (Supplementary Table 6)." The statement does not seem to be true at all. I am not sure what items the authors are trying to compare. At least the H₂ evolution rate of [Mo₃S₁₃]@ZnP-Pz-PEO-COF (10.7 mmol/(g·h)) does not surpass many listed items in Table S6, including MoS₁₃@EB-COF (13.2 mmol/(g·h)), Tp-2C/BPy₂+COF (34.6 mmol/(g·h)), TtaTfa (20.7 mmol/(g·h)), NKCOF-108 (11.6 mmol/(g·h))

Reviewer #3 (Remarks to the Author):

Jiang et.al. constructed different photocatalytic systems using a different molecular interfaces. Their results are very interesting and the paper can be published after a major revision Here are my comments:

1- It was not very clear to me the way that they estimate/calculate the exciton binding energy from the temperature-dependent PL measurements.

2- It is very hard for me to understand the kinetic traces in Fig. 6. I do not understand what decay means after around 50 ps.

3- All the figures need to be edited to be able to see the Legends.

4- One of the papers that proves the COF application in the photocatalytic application is here and it will be great if the author mentions it (<https://doi.org/10.1038/s41467-022-28409-2>). This study covers the charge dynamics of similar COF materials in both the Visible and IR ranges.

Point-to-Point Answers to Comments

Reviewer #1 (Remarks to the Author): All the concerns raised by the reviewers have been adequately addressed. The addition of ultrafast experiments is also a plus that is consistent with the photocatalytic results. The manuscript can be publishable provided that the authors address the following last comments.

1. P. 13, line 335, and fig. 6f. The authors describe the substantial difference between the transient absorption of the COF and the COF+catalyst. However, no detailed explanation is given as to the featuring spectral signatures observed in the latter system (fig. 6f). These spectral features should be properly assigned in order to confirm the observed behavior and compare the corresponding decay kinetics.

Answer: We appreciate your kind suggestions. We have revised the paragraph with new sentences and details for fig. 6e-g as “ Fig. 6f shows the dramatical change of the TA spectra with a new positive peak around 480 nm when upon adding the $[\text{Mo}_3\text{S}_{13}]^{2-}$ to the ZnP-Pz-PEO-COF. This new positive excited-state absorption band originates from the reduction of $[\text{Mo}_3\text{S}_{13}]^{2-}$ to $[\text{Mo}_3\text{S}_{13}]^{3-}$ species by the photoinduced electron. Therefore, the kinetics of ZnP-Pz-PEO-COF probed at 500 nm were calculated to show the average life time (τ) and rate constant (k) as 24 ps and 0.41 ns^{-1} , respectively (Fig. 6h, blue curve, Supplementary Table 7).”, for a better understanding.

2. The temperature dependent emission results and the ultrafast experiments on pp. 12,13 should be described on the "Results" section. A proper "Discussion" section should be added which discusses in a more comprehensive manner the results previously outlined.

Answer: We appreciate your kind suggestions. The temperature dependent emission results and the ultrafast experiments have been removed to the “Exciton binding energy” and “Femtosecond transient absorption and electron dynamics” sections of “Results”, respectively. We have added a “Discussion” section to explain the comprehensive results.

We appreciate this reviewer for suggestive comments to improve the quality.

Reviewer #2 (Remarks to the Author): I appreciate the authors' efforts on addressing the concerns and questions that were raised earlier. There are still missing details of this research and I am not convinced that the current quality and novelty of the work warrant the acceptance of Nat. Commun.

1. Novelty issue. Multivariate COFs (Polym. Chem., 2021,12, 3250) and Mo-cluster encapsulated COFs (Chem. Commun., 2018, 54, 13563; ACS Energy Lett. 2018, 3, 400) have been both evaluated for hydrogen production as photocatalysts. The hydrophilic motif was also investigated from the previous ref (Angew. Chem. Int. Ed., 2019, 58, 18290). This work integrates all the three components in one system for photocatalytic H₂ production and is attractive to certain audience. It seems that a more specific journal would be an appropriate venue for the publication.

Answer: We appreciate your kind suggestions. The photocatalytic H₂ evolution reactions are complicated which involve the photoexcitation, separation and transportation of photo-generated charge carriers and redox reactions. However, none of the above papers have clearly addressed the necessary of interfacial design for photocatalytic COFs. In this paper, we elucidate the essential interfaces for photoinduced hydrogen evolution from water with non-noble metal catalytic centres and unambiguously reveal the necessary structures for photocatalytic reactions. By setting the structural parameters on skeleton for electron flow, coordination sites for ligating metal centre and nanochannels for mass transport, we predesigned new structures for each process and integrated them into one framework material. This concept and the tailor-made materials are both novel. Under this original scheme, we successfully developed an integrated interfacial design strategy to prepare COF photocatalysts, which can merge three distinguished interfaces in a single material: (1) the π -electronic interface for the light-harvesting and electron transfer; (2) the immobilisation interface for the ligation of non-noble metal redox centres; (3) the mass transport interface for the water delivery. These multi functions were difficult to achieve simultaneously in the above papers and other kind of materials. We believe that our approach will open the way to develop integrated heterogeneous photocatalysts for the actionable solar-to-chemical energy conversion and green fuel production, as agreed by other reviewers.

2. "The small peak at 4.0° might be attributed to the formation of some extended slipped J-aggregate stacks, which was observed in other porphyrin-based COFs.³⁰" This explanation is not satisfactory and I am not able to identify any 4.0° discussion in the cited reference.

Answer: We appreciate your kind suggestions. In reference 30 (J. Am. Chem. Soc. 2018, 140, 16544–16552), there is no discussion for the peak at 4.0° due to the different crystal size between ZnP-Pz-DHTP-COF (this work) and TT-Por COF (Ref). However, the JACS authors found that two shoulders (3.4° and 6.8°) appeared on the 100 and 200 reflections. They explained it as "**Both a shoulder on the 100 reflection and a splitting of the 200 reflection into two new distinct peaks indicate a loss in symmetry of the unit cell. Optimizing the geometry, including all unit cell parameters and using a periodic force-field treatment resulted in a simulated pattern that matches the experimental one. In order to produce the correct pattern without neglecting the geometry of the building blocks, the unit cell parameter γ must deviate from 90°, resulting in an offset of the adjacent layers (Figure 2a,b). This offset results in a staircase-like stacking behavior, which is prevalent in one direction and can be distinctly recognized in the PXRD pattern (Figure 2a).**" They also used force-field-based simulated annealing method to simulate the structure of TT-Por-COF with a periodic system of 1000 unit cells. The result is that "**This procedure resulted in significantly distorted layers, which, however, retained the staircase**

shifted pattern. PXRD patterns predicted for idealized and annealed supercells (see the SI for details) fit the experimental curve very well (Figure 3).” As a result, we think that the shoulder peak at 4.0° near to the 100 reflection (3.4°) might be attributed to the formation of some slipped stackings. The shoulder peaks are also observed in many other porphyrin COFs published in *J. Am. Chem. Soc.* 2020, 142, 16723–16731; *J. Am. Chem. Soc.* 2020, 142, 49, 20763–20771; and *J. Am. Chem. Soc.* 2018, 140, 1116–1122.”

3. “We can confirm that the ratio of Pz and DHTP is 1 to 1, so we arrange the two units on the para position of COF structures for a better understanding.” Has the resultant 1:1 ratio between Pz and DHTP been experimentally confirmed, though the initial experimental loading was 1:1? I didn’t this experimental evidence in neither manuscript nor SI. A better understanding does not come from the arbitrary assignment of chemical species in the space without any explanation. An explanation or illustration for this arbitrary para-arrangement has to be added to the figure caption.

Answer: We appreciate your kind suggestions. We have done additional experiments and conducted the solution-state ^1H NMR spectroscopy of the hydrolyzed ZnP-Pz-DHTP-COF to confirm the molar ratio of Pz and DHTP. We revised the manuscript by adding a new sentence “The ^1H NMR spectroscopy of hydrolyzed ZnP-Pz-DHTP-COF revealed that the ratio of PzDA and DHTA was 1/1, confirming the successful synthesis of three-component COFs (Supplementary Fig. 1).”.

4. The resolution of Figure 1 has not been improved, as pointed by the other referee.

Answer: We appreciate your kind suggestions. The resolution of Figure 1 was revised and improved.

5. Figure 3, EDX elemental mapping for Zn was not clearly illustrated, probably due to the demetallation during the synthesis. Given the stability of Zn-base porphyrin has relatively low tolerance toward protons, I am a little concerned by the demetallation of the porphyrin units during the synthesis (exposed to HOAc). Any experimental data to disprove the demetallation? Is the role of Zn critical for light harvesting purpose?

Answer: We appreciate your kind suggestions. We have done additional experiments and determined the content of Zn in ZnP-Pz-DHTP-COF. The content of Zn was 6.54 wt%, which was close to the theoretical value of 6.62 wt%. The results confirmed the stability of Zn-porphyrin under solvothermal conditions. We revised the manuscript by adding a new sentence “The content of Zn for ZnP-Pz-DHTP-COF was determined to be 6.54 wt% by inductively coupled plasma optical emission spectroscopy (ICP-OES), which was close to the theoretical value (6.62 wt%), revealing the stability of Zn species under the solvothermal conditions.”. Moreover, we performed the synthetic experiment of ZnP-Pz-DHTP-COF in the absence of aldehyde monomers. The solvent was removed under vacuum and the resulting solid was washed with H_2O . The MALDI-TOF (shown below) results showed that no free-base porphyrin appeared in the resulting solid under the synthetic conditions of COFs. As shared in the porphyrin field, demetallation requires the use of strong HCl condition.

6. Page 4, line 90. “The hydrophilic ZnP-Pz-PEO-COF with pyrazine sites was synthesised in 96% yield...” How has this reaction yield been determined? Please be specific. This number is slightly different from data on Page 6, line 146, which only indicate 93% of -OH groups involve in the reaction.

Answer: We appreciate your kind suggestions. The elemental analysis results showed that 93% of -OH transformed to PEO chain. We calculated the yield of ZnP-Pz-PEO-COF based on the molecular weight of repeated units for ZnP-Pz-DHTP_{0.07}-PEO_{0.93} to be a 96% yield. We revised the manuscript by adding a new sentence “The powder was collected and dried at room temperature under vacuum overnight to give ZnP-Pz-PEO-COF in an isolated yield of 96% (based on the 93% of PEO and 7% of -OH units which were determined by elemental analysis).”.

7. Page 10, line 256, “Remarkably, [Mo₃S₁₃]²⁻@ZnP-Pz-PEO-COF is much superior to those of the state-of-the-art Pt-based systems (Supplementary Table 6).” The statement does not seem to be true at all. I am not sure what items the authors are trying to compare. At least the H₂ evolution rate of [Mo₃S₁₃]²⁻@ZnP-Pz-PEO-COF (10.7 mmol/(g·h)) does not surpass many listed items in Table S6, including MoS₁₃@EB-COF (13.2 mmol/(g·h)), Tp-2C/BPy₂⁺-COF (34.6 mmol/(g·h)), TtaTfa (20.7 mmol/(g·h)), NKCOF-108 (11.6 mmol/(g·h))

Answer: We appreciate your kind suggestions. We revised it to “Remarkably, [Mo₃S₁₃]²⁻@ZnP-Pz-PEO-COF is comparable to or even higher than those of the state-of-the-art Pt-based systems (Supplementary Table 6).”, for a better understanding.

We appreciate this reviewer for suggestive comments to improve the quality.

Reviewer #3 (Remarks to the Author): Jiang et.al. constructed different photocatalytic systems using a different molecular interfaces. Their results are very interesting and the paper can be published after a major revision. Here are my comments:

1. It was not very clear to me the way that they estimate/calculate the exciton binding energy from the temperature-dependent PL measurements.

Answer: We appreciate your kind suggestions. The details for the calculation of exciton binding energy from the temperature-dependent PL measurements have been provided in the supporting information.

“Temperature-dependent photoluminescence spectra were recorded by Edinburgh Instruments (FLS980) to determine the exciton binding energy (E_b) of COFs. The intensity of PL decreases when the temperature increases. The corresponding E_b is calculated through fitting the intensity data with Arrhenius equation, $I(T)=I_0/(1+A\exp(-E_b/k_B T))$.^{7”}

2. It is very hard for me to understand the kinetic traces in Fig. 6. I do not understand what decay means after around 50 ps.

Answer: We appreciate your kind suggestions. In order to present the most useful information for the kinetic traces located before 10 ps, we conducted a zoom in operation for the kinetic curves in fig. 6g-i. The full kinetic curves of ZnP-Pz-PEO-COF and $[\text{Mo}_3\text{S}_{13}]^{2-}$ @ZnP-Pz-PEO-COF without zoom in operation presented below. It is quite difficult to achieve useful information for both samples after 50 ps at this stage.

3. All the figures need to be edited to be able to see the Legends.

Answer: We appreciate your kind suggestions. The legends of all figures were enlarged and the resolution of figures were improved.

4. One of the papers that proves the COF application in the photocatalytic application is here and it will be great if the author mentions it (<https://doi.org/10.1038/s41467-022-28409-2><<https://ddec1-0-en-ctp.trendmicro.com:443/wis/cl icktime/v1/query?url=https%3a%2f%2fdoi.org%2f10.1038%2fs41467%2d022%2d28409%2d2&umid=789cd9e8-2db2-4da3-b0ed-d8b26de2ec5d&auth=8d3ccd473d52f326e51c0f75cb32c9541898e5d5-2f9848926fb0ad2cf24cd9b8f1079952c7ee4414>>). This study covers the charge dynamics of similar COF materials in both the Visible and IR ranges.

Answer: We appreciate your kind suggestions. The paper (Nat. Commun. 2022, 13, 845.) was added as ref 36.

We appreciate this reviewer for suggestive comments to improve the quality.

REVIEWER COMMENTS

Reviewer #1 (Remarks to the Author):

I generally agree with the authors that their manuscript adds an important piece of work to the current literature on materials for photochemical hydrogen evolution and, as such, it may deserve publication in Nature Commun.

While most of the comments have been adequately addressed, I do not feel that my previous concern related to the new ultrafast spectroscopy experiments has been properly considered. In particular, the authors attribute the new spectral feature at 480 nm in the COF+catalyst system to the reduced catalyst. While I might agree that the spectral changes are considerably different when comparing COF and COF+catalyst samples, according to the expected effect of the catalyst, the attribution of the new absorption pattern to the reduced catalyst must be properly confirmed. I would suggest that the authors check for possible literature or confirm their attribution by spectroelectrochemical measurements.

Reviewer #2 (Remarks to the Author):

The authors have fully addressed my technical concerns raised earlier. Thank you for the additional efforts.

Reviewer #3 (Remarks to the Author):

I see that the author addressed my comments/questions and it looks very suitable for publication in Nat. comm.

Point-to-Point Answers to Comments

Reviewer #1 (Remarks to the Author):

I generally agree with the authors that their manuscript adds an important piece of work to the current literature on materials for photochemical hydrogen evolution and, as such, it may deserve publication in Nature Commun.

While most of the comments have been adequately addressed, I do not feel that my previous concern related to the new ultrafast spectroscopy experiments has been properly considered. In particular, the authors attribute the new spectral feature at 480 nm in the COF+catalyst system to the reduced catalyst. While I might agree that the spectral changes are considerably different when comparing COF and COF+catalyst samples, according to the expected effect of the catalyst, the attribution of the new absorption pattern to the reduced catalyst must be properly confirmed. I would suggest that the authors check for possible literature or confirm their attribution by spectroelectrochemical measurements.

Answer: We appreciate your kind suggestions.

Firstly, we have electrolyzed $(\text{NH}_4)_2\text{Mo}_3\text{S}_{13}$ in CH_3CN (0.1M $t\text{-Bu}_4\text{NPF}_6$) at -0.75V for 1h under Ar, then we checked the UV-Vis absorption of $(\text{NH}_4)_2\text{Mo}_3\text{S}_{13}$ before and after electrolysis, the intensity of absorption from 400 to 500 nm was enhanced after electrolysis (supplementary figure 21), which is close to the position of the absolved band at 480 nm. Secondly, we have performed the fs-TA spectrum of $(\text{NH}_4)_2\text{Mo}_3\text{S}_{13}$ to show a broad positive GSB peak from 500-750 nm as supplementary figure 20.

Based on these two results, the peak of $[\text{Mo}_3\text{S}_{13}]^{2-}@ZnP\text{-Pz-PEO-COF}$ at 480 nm might originate from the reductive intermediate by the photoinduced electron. We have revised the statement as “This new positive excited-state absorption band might originate from the reductive intermediate by the photoinduced electron compared to the fs-TA results of ZnP-Pz-PEO-COF (Fig. 6e) and $(\text{NH}_4)_2\text{Mo}_3\text{S}_{13}$ (Supplementary Fig. 20). Moreover, the UV-Vis spectrum of $(\text{NH}_4)_2\text{Mo}_3\text{S}_{13}$ in CH_3CN (0.1M $t\text{-Bu}_4\text{NPF}_6$) presented an enhancement of the absorption from 400 to 500 nm after 1h electrolysis at -0.75V under Ar, further suggesting the new peak of TA spectrum originated from the reductive intermediate (Supplementary Fig. 21).” for a better understanding.

We appreciate this reviewer for suggestive comments to improve the quality.

Reviewer #2 (Remarks to the Author):

The authors have fully addressed my technical concerns raised earlier. Thank you for the additional efforts.

We appreciate this reviewer's recommendation.

Reviewer #3 (Remarks to the Author):

I see that the author addressed my comments/questions and it looks very suitable for publication in Nat. comm.

We appreciate this reviewer's recommendation.

REVIEWERS' COMMENTS

Reviewer #1 (Remarks to the Author):

The previous concern has been adequately addressed by the authors. The manuscript may deserve publication.

Point-to-Point Answers to Comments

Reviewer #1 (Remarks to the Author):

The previous concern has been adequately addressed by the authors. The manuscript may deserve publication.

We appreciate this reviewer's recommendation.

Reviewer #3 (Remarks to the Author):

I see that the author addressed my comments/questions and it looks very suitable for publication in Nat. comm.

We appreciate this reviewer's recommendation.